# Reciprocal rescue of Wolfram syndrome by two causative genes

Su Jin Ham [1,2,3], Eunju Yoon [1,2,3], Da Hyun Lee [1,2,3], Sehyeon Kim [1,2], Heesuk Yoo [1,2] & Jongkyeong Chung [1,2 ✉]

## Abstract

**Wolfram syndrome (WS) is marked by juvenile-onset diabetes mellitus, optic atrophy, diabetes insipidus, and sensorineural hearing loss. The causative genes, *WFS1* and *CISD2*, correspond to WS types 1 and 2, respectively. Here, we establish their mutual indispensability for inositol 1,4,5-triphosphate receptor (IP$_3$R) activity, demonstrating their ability to restore reduced IP$_3$R activity in WFS1- or CISD2-deficient mammalian cells. Additionally, our *Drosophila* WS models lacking *dWFS1* or *dCISD* exhibit diabetes-like phenotypes analogous to WS patients, and overexpression of *dWFS1* and *dCISD* in the flies alleviates their phenotypes. We have engineered a peptide containing the CDGSH domain of CISD2, critical for its interaction with IP$_3$R. Overexpression of our CISD2 peptide or treatment with its cell-penetrating peptide (CPP)-conjugated form restores calcium homeostasis in WFS1- or CISD2-deficient cells, and overexpressing the homologous dCISD peptide suppresses diabetes-like phenotypes in WS model flies. These findings underscore the intricate involvements of WFS1 and CISD2 in ER calcium regulation and provide potential therapeutic prospects for WS-related diabetes.**

**Keywords** Calcium; CISD2; IP$_3$R; WFS1; Wolfram Syndrome
**Subject Categories** Membranes & Trafficking; Metabolism; Molecular Biology of Disease

## Introduction

Wolfram syndrome (WS) is a rare genetic disorder that is characterized by diabetes insipidus (DI), childhood-onset diabetes mellitus (DM), optic atrophy (OA), and deafness (D) (collectively abbreviated as DIDMOAD) (Scolding et al, 1996; Urano, 2016) and also presents an array of complex symptoms involving bladder, bowel, and temperature regulation issues, alongside diverse endocrinological, psychiatric, and neurological abnormalities (Ito et al, 2007; Pallotta et al, 2019; Urano, 2016). Typically, juvenile diabetes mellitus is the first symptom to manifest, followed by optic

nerve atrophy and central diabetes insipidus, in addition to significant urinary tract complications that affect most WS patients and result in obstructions and difficulties in controlling urine flow (Annamalai et al, 2019; Pallotta et al, 2019; Urano, 2016). Neurological manifestations such as ataxia and brain stem atrophy often lead to severe complications like central apnea (Aldenhovel et al, 1991; Ito et al, 2007). Despite the multifaceted clinical presentation of WS, there exists an absence of effective therapies to halt its progression, highlighting an urgent need for the development of innovative treatment strategies.

Mutations in the genes *Wolfram syndrome 1* (*WFS1*) and *CDGSH iron sulfur domain 2* (*CISD2*), also termed *NAF-1*, *Miner1*, and *WFS2*, are known to cause WS types 1 (WS1) and 2 (WS2), respectively, which result in clinical manifestations that slightly vary (Amr et al, 2007; Gaines et al, 2021; Inoue et al, 1998; Karmi et al, 2017; Strom et al, 1998; Wiley et al, 2007). WS1, the more prevalent form, primarily involves recessive *WFS1* mutations impacting the protein wolframin (WFS1) that is widely expressed across various tissues and integral for multiple cellular functions, notably at mitochondria-associated membranes (MAMs) (Delprat et al, 2018; Rigoli et al, 2022; Strom et al, 1998). Although predominantly recessive, a small subset of WS1 patients harbors dominant *WFS1* mutations that are often associated with milder phenotypes characterized by optic atrophy and sensorineural deafness (Cryns et al, 2003; Inoue et al, 1998). WS2, a rarer manifestation that has been linked to recessive *CISD2* mutations (Amr et al, 2007), shares many clinical features with WS1 including diabetes mellitus, but distinctively presents with gastrointestinal complications like peptic ulcers and bleeding tendencies without diabetes insipidus (Al-Sheyyab et al, 2001; Rosanio et al, 2022; Urano, 2016).

A comprehensive array of in vivo and in vitro models has been pivotal in elucidating the intricate molecular mechanisms underpinning WS. Investigations using fibroblasts from patients carrying recessive *WFS1* mutations showcased disrupted mitochondrial function and impaired calcium homeostasis (Angebault et al, 2018; La Morgia et al, 2020), signifying heightened calcium flux from the endoplasmic reticulum (ER) to the mitochondria. Studies with primary cortical neurons from WFS1-deficient mice revealed ER stress induction, resulting in dysfunctional inositol 1,4,5-triphosphate receptor (IP$_3$R) and perturbed calcium equilibrium accompanied by altered mitochondrial dynamics, fusion inhibition,

[1]Institute of Molecular Biology and Genetics, Seoul National University, Seoul 08826, Republic of Korea. [2]School of Biological Sciences, Seoul National University, Seoul 08826, Republic of Korea. [3]These authors contributed equally: Su Jin Ham, Eunju Yoon, Da Hyun Lee. ✉E-mail: jkc@snu.ac.kr

aberrant mitochondrial trafficking, and increased mitophagy (Cagalinec et al, 2016). Contrastingly, limited research on WS2 models hampers the understanding of CISD2's functional implications in the disease. Studies examining CISD2's role in neuronal cells inferred its direct involvement in regulating intracellular calcium levels (Chang et al, 2010; Chen et al, 2009; Rouzier et al, 2017; Wiley et al, 2013). Moreover, murine models with CISD2 mutations exhibited skeletal and neuronal degeneration, dysregulated calcium homeostasis, increased autophagy, and mitochondrial dysfunction (Chen et al, 2009; Wu et al, 2012), resembling some pathophysiological characteristics observed in WFS1 mutant rodents (Akiyama et al, 2009; Plaas et al, 2017; Richard et al, 2023). CISD2 has also been reported to directly interact with B cell lymphoma 2 (BCL2) on the ER membrane (Chang et al, 2012a; Chang et al, 2010; Chang et al, 2012b). Interestingly, it seems that the CISD2-BCL2 complex does not regulate caspase-dependent apoptosis, but rather participates in the BCL2-regulated autophagy pathway via a physical association with IP$_3$R that in turn modulates its calcium channel activity and reduces the ER calcium store (Chang et al, 2010).

As previous studies have suggested dysfunctional ER calcium homeostasis as one of the causes of WS pathogenesis, it is crucial to understand the mechanisms that regulate IP$_3$R, a calcium channel predominantly localizing to the ER membrane (Callens et al, 2022; Delprat et al, 2018; Mishra et al, 2021). IP$_3$R is responsible for liberating calcium ions from the ER into the cytosol and mitochondria in response to various cues, and the released calcium assumes a critical role as a secondary messenger indispensable for numerous cellular functions (Hamada and Mikoshiba, 2020). Furthermore, IP$_3$R operates as an intricate signaling center that processes a spectrum of cellular inputs including inositol 1,4,5-trisphosphate (IP$_3$), calcium, ATP, thiol modifications, and phosphorylation (Hamada and Mikoshiba, 2020). The intricate interplay among these inputs yields precisely timed and spatially defined cytosolic and mitochondrial calcium signals. Both IP$_3$ and calcium act as primary agonists for IP$_3$R channel activation, with both elements being indispensable (Hamada and Mikoshiba, 2020). Additionally, various proteins such as BCL2 (Chang et al, 2010; Rong et al, 2008), B cell lymphoma-extra large (BCL-xL) (Rosa et al, 2022), myeloid leukemia 1 (Mcl1) (Eckenrode et al, 2010), Beclin1 (Vicencio et al, 2009), Annexin A1 (Vais et al, 2020), IP$_3$R-binding protein released with IP$_3$ (IRBIT) (Ando et al, 2006), neuronal calcium sensor 1 (NCS1) (Angebault et al, 2018), and ER protein sigma-1 receptor (Sig-1R) (Hayashi and Su, 2007) interact with IP$_3$R and modulate its function. These characteristics allow IP$_3$R to assimilate signals from diverse pathways and grant its crucial role in regulating various cellular processes including apoptosis, autophagy, and ER stress response (Hamada and Mikoshiba, 2020). Here, we demonstrate the direct interaction between WFS1 and CISD2 with IP$_3$R and propose a novel IP$_3$R regulator that may be employed for treatment purposes.

In this study, we sought to gain deeper insights on the roles of WFS1 and CISD2 in ER calcium homeostasis and how deficits in either result in WS pathogenesis. Through our experiments, we confirmed that both WFS1 and CISD2 bind directly to and activate IP$_3$R to stimulate ER calcium release. In addition, we generated in vivo *Drosophila* models of WS by generating mutants of fly homologs of human *WFS1* and *CISD2*, *dWFS1*, and *dCISD*, respectively, and observed diabetes-like phenotypes that resemble

major human WS symptoms that were rescued by IP$_3$R overexpression, demonstrating that restoring calcium homeostasis can alleviate WS-related diabetic complications. We also discovered that the expression of either *WFS1* or *CISD2*, or *dWFS1* or *dCISD* in flies, ameliorates the disrupted calcium homeostasis and diabetic phenotypes caused by the deficit of either gene. Lastly, our study identified a short CISD2 peptide of 18 amino acids, and the respective dCISD peptide in flies, that effectively binds to and activates IP$_3$R to ameliorate diabetes-like symptoms, presenting a novel therapeutic method to treat WS.

## Results

### Loss of *WFS1* or *CISD2* leads to abnormal IP$_3$R activity and Wolfram syndrome-related diabetes-like phenotypes in *Drosophila*

To observe the roles of WFS1 and CISD2 in intracellular calcium homeostasis, we generated WFS1- or CISD2-deficient human embryonic kidney (HEK) 293 cells using the clustered regularly interspaced short palindromic repeats (CRISPR)-Cas9 system (Appendix Fig. S1A–C). We utilized an ER-specific green-calcium-measuring organelle-entrapped protein indicator, G-CEPIA1er, to visualize ER calcium flux in cells. We first measured ER calcium release under ATP treatment which induces production of inositol 1,4,5-trisphosphate (IP$_3$) by stimulating phospholipase C (PLC) via binding to G-protein-coupled receptor (GPCR) (Suzuki et al, 2014). As previously reported (Liiv et al, 2024), ER calcium release was decreased both in WFS1 and CISD2 knockout cells in comparison to wild type (WT) control cells (Fig. 1A). We examined cytosolic calcium levels as well using a red-shifted genetically-encoded calcium indicator, RCaMP1h, as a calcium indicator under ATP treatment (Hirabayashi et al, 2017). In accordance with our ER calcium results, cytosolic calcium levels were also reduced in both WFS1 and CISD2 knockout cells (Fig. 1B). To determine which calcium channel was responsible for the reduced ER calcium release and cytosolic calcium levels exhibited in mutant cells, we conducted a follow-up experiment that discerns ER calcium influx and efflux measurements. We permeabilized cells with escin and perfused them with calcium chloride to observe ER calcium influx through sarco/endoplasmic reticulum Ca$^{2+}$-ATPase (SERCA), which is an ER calcium transporter that pumps calcium into the ER from the cytosol. Afterwards, cells were exposed to a buffer containing IP$_3$ to activate IP$_3$ receptor (IP$_3$R), an ER calcium channel through which ER luminal calcium is released into the cytosol. Neither WFS1 nor CISD2 knockout cells showed ER calcium uptake that varies from control cells, however, ER calcium release triggered by IP$_3$ was significantly decreased in both WFS1 and CISD2 lacking cells compared to controls (Fig. 1C). Altogether, these results indicate that both WFS1 and CISD2 manipulate IP$_3$R activity such that the loss of either reduces IP$_3$R activity.

To test the roles of WS causative genes in vivo, we created models of WS in *Drosophila* by employing mutants of the fly homologs of *WFS1* and *CISD2*, *dWFS1*, and *dCISD*, respectively. *dCISD* mutant flies were generated in a previous study (Ham et al, 2023), and *dWFS1* mutants were generated using the CRISPR-Cas9 system (Appendix Fig. S1D,E). Surprisingly, both *dWFS1* and *dCISD* mutant flies were lighter in weight when aged to 30 days of age even though feeding intake did not differ from control *w1118*

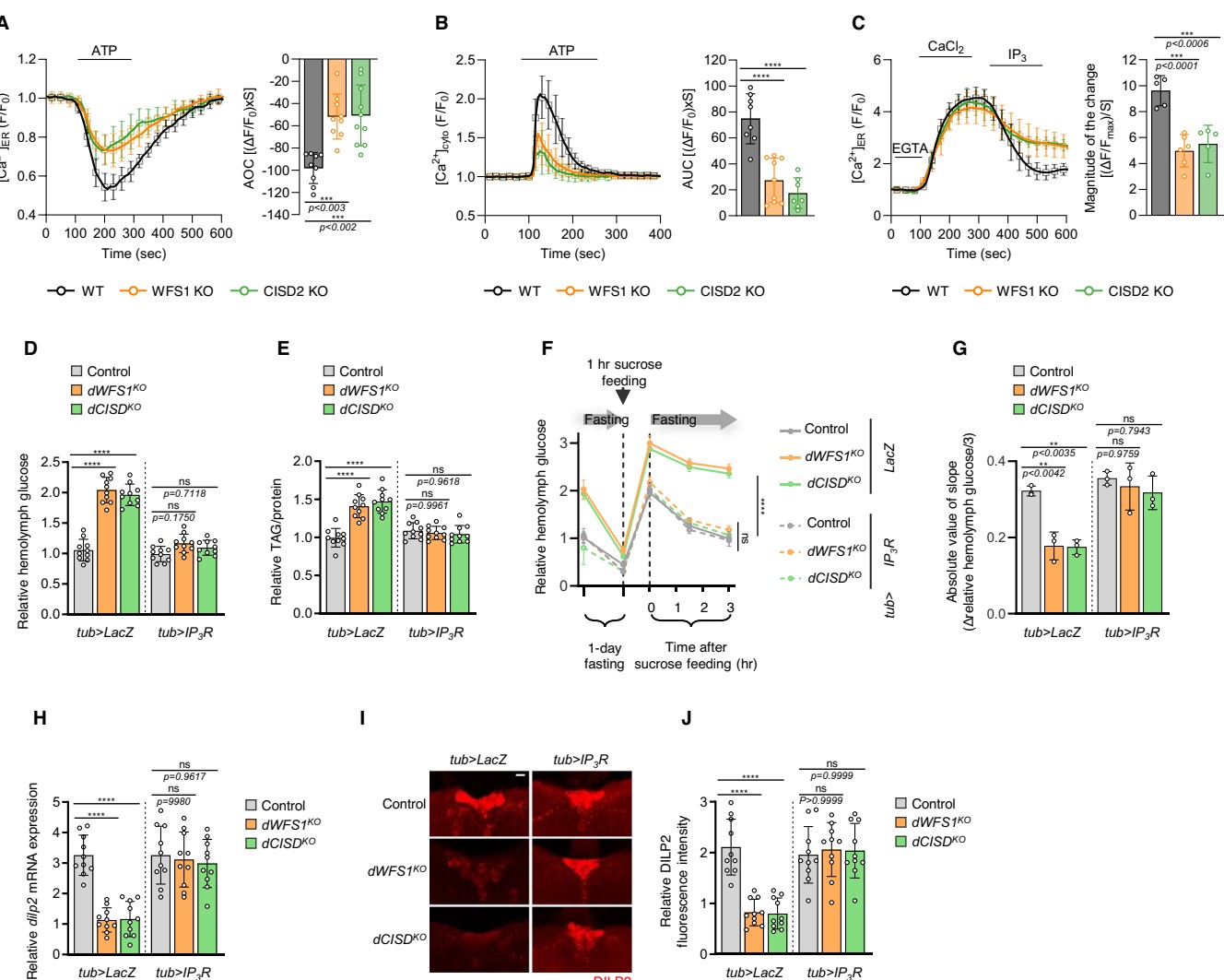

**Figure 1. Loss of WFS1 and CISD2 exhibit decreased IP₃R activity, and diabetes-like phenotypes of *dWFS1* and *dCISD* mutant flies are ameliorated by restoring IP₃R activity.**

(A) Measurement of ER calcium flux in WT (black, $n = 115$ cells, 9 coverslips), WFS1 KO (orange, $n = 110$ cells, 9 coverslips), and CISD2 KO (green, $n = 83$ cells, 10 coverslips) HEK293 cells. (B) Measurement of cytosolic calcium flux in WT (black, $n = 71$ cells, 8 coverslips), WFS1 KO (orange, $n = 69$ cells, 9 coverslips), and CISD2 KO (green, $n = 59$ cells, 7 coverslips) HEK293 cells. 100 μM ATP was treated to initiate IP₃R-mediated ER calcium release. The right-side bar graphs indicate the quantification of the normalized calcium traces using area-over-the-curve (AOC) or area-under-the-curve (AUC) of calcium fluxes during ATP treatment. (C) Measurement of IP₃R activity in WT (black, $n = 95$ cells, 5 coverslips), WFS1 KO (orange, $n = 74$ cells, 6 coverslips), and CISD2 KO (green, $n = 86$ cells, 5 coverslips) HEK293 cells. 40 μM β-escin was treated to permeabilize cells for 100 s. The cells were then washed with intracellular medium (ICM) for 5 min. 0.65 mM CaCl₂ was delivered to induce the influx of ER calcium. When a steady state was reached, 0.5 μM IP₃ was introduced to induce ER calcium release through IP₃R. The right-side bar graphs show the magnitude of the change during IP₃ treatment. (D) Relative levels of hemolymph glucose from 30-day-old *w1118*, *dWFS1* KO, or *dCISD* KO flies expressing either the transgenic control UAS-*LacZ* or UAS-*Itpr*, the fly IP₃R homolog, with the *tub-Gal4* driver. Data were normalized to *tub>LacZ* controls ($n = 10$). (E) Relative levels of TAG divided by their respective protein levels of 30-day-old flies, normalized to *tub>LacZ* flies ($n = 10$). (F) Relative levels of hemolymph glucose of 30-day-old flies followed through the glucose tolerance test, normalized to *tub>LacZ* flies ($n = 3$). ns indicates *p*-values of 0.6602 (control, *tub>LacZ* vs. *dWFS1* KO, *tub>IP₃R*) and 0.9876 (control, *tub>LacZ* vs. *dCISD* KO, *tub>IP₃R*). (G) The absolute value of the slope after the 1 h sucrose feeding to quantify the recovery of heightened glucose levels ($n = 3$). (H) Relative *dilp2* mRNA levels in whole fly bodies as quantified through real-time PCR, normalized to *dWFS1* KO, *tub>LacZ* flies ($n = 10$). (I) Staining of fly brains using anti-DILP2 antibody ($n = 10$). Scale bar, 10 μm. (J) Quantifications of the fluorescence intensity of DILP2 brain images, normalized to *dCISD* KO, *tub>LacZ* flies ($n = 10$). Data information: All figures are representatives of three or more independent experiments. All quantifications were analyzed by one-way ANOVA with Tukey multiple-comparison test. **$p < 0.01$, ***$p < 0.001$, ****$p < 0.0001$. ns, not significant. All data are presented as mean ± SD. Source data are available online for this figure.

flies (Appendix Fig. S2A,B). As these results are analogous to the weight loss experienced by patients with type 1 diabetes and diabetes mellitus is a major WS symptom, we examined WS flies for additional diabetes-related phenotypes (Chatterjee et al, 2017; Conway et al, 2009; Ghirardello et al, 2014). Strikingly, we discovered that *dWFS1* and *dCISD* mutants have hemolymph glucose levels that increase with age, reaching roughly double of that of controls by 30 days of age (Appendix Fig. S2C). Mutant flies also exhibit elevated triacylglycerol (TAG) levels when aged (Appendix Fig. S2D). Furthermore, 30-day-old flies lacking *dWFS1*

or *dCISD* had lower mRNA levels of *dilp2* and the corresponding DILP2 protein, akin to human insulin (Appendix Fig. S2E–G). These results demonstrate that aged *dWFS1* and *dCISD* mutant flies exhibit diabetes-like phenotypes that resemble the diabetic complications that human WS patients experience, firmly establishing these mutant lines as valid *Drosophila* WS models. We also confirmed that deficits in *dWFS1* and *dCISD* do not induce developmental defects, with larvae showing developmental timelines and sizes similar to controls, reaching pupariation by 8 days after egg laying (Appendix Fig. S3A,B). Since the diabetes-related phenotypes that manifest in our aged *Drosophila* models soundly mimic the major diabetic complications that WS patients present, we focused on diabetic phenotypes and employed 30-day-old flies for the remainder of our study.

As we found diminished IP$_3$R activity in *WFS1* and *CISD2* mutant cell lines, we tested if loss of *dWFS1* or *dCISD* induces impaired calcium homeostasis in *Drosophila*. We employed flies expressing ERGCaMP6-210, an ER calcium indicator, or GCaMP5G, a cytosolic calcium indicator. Then, we crossed them with *dWFS1* or *dCISD* mutant flies and measured ER and cytosolic calcium fluxes. Consistent with our cell experiments, *dWFS1* and *dCISD* null flies exhibited decreased ER calcium release and cytosolic calcium levels compared to control flies (Appendix Fig. S4A–D). Furthermore, we investigated if overexpression of *IP$_3$R* could rescue the defected calcium fluxes in *dWFS1* and *dCISD* mutant flies. We used the ubiquitous *tubulin (tub)-Gal4* driver and UAS-*Itpr*, the fly homolog of human *IP$_3$R*, to overexpress *IP$_3$R* in flies. When crossed with *IP$_3$R* transgenic flies, reduced ER calcium release and cytosolic calcium levels in *dWFS1* or *dCISD* mutant flies were recovered (Appendix Fig. S4A–D). Additionally, crossing *IP$_3$R* transgenic flies with control flies resulted in increased ER calcium release and cytosolic calcium levels compared to control flies (Appendix Fig. S4A–D). Consistent with *Drosophila* calcium measurements, overexpression of IP$_3$R1 in HEK293 WT cells increased ER calcium release and cytosolic calcium levels (Appendix Fig. S5A–D), as expected. When IP$_3$R1 was overexpressed in WFS1 lacking cells, the reduced ER calcium release and cytosolic calcium levels were rescued (Appendix Fig. S5A,B). Similarly, overexpression of IP$_3$R1 restored the diminished ER calcium release and cytosolic calcium levels in CISD2 lacking cells (Appendix Fig. S5C,D).

As we confirmed that increasing *IP$_3$R* expression rescues disrupted calcium homeostasis in *dWFS1* and *dCISD* null flies and mammalian cells, we studied whether overexpression of *IP$_3$R* could rescue the diabetes-like phenotypes exhibited by *dWFS1* and *dCISD* mutant flies. Overexpression of *IP$_3$R* ameliorated the diabetes-like phenotypes exhibited by both *dWFS1* and *dCISD* mutant flies, including elevated hemolymph glucose and TAG levels (Fig. 1D,E). In addition, we performed a modified version of glucose tolerance test (GTT) in which 30-day-old flies were fasted for 24 h, fed 10% sucrose water for 1 h, then fasted again, during which hemolymph glucose levels were followed. *dWFS1* and *dCISD* mutant flies exhibited sharper increases in hemolymph glucose levels after the brief sucrose feeding followed by significantly slower recovery of such heightened glucose levels, demonstrating abnormal GTT responses, but *IP$_3$R* overexpression restored these abnormal GTT responses (Fig. 1F,G). *IP$_3$R* overexpression also rescued the lower *dilp2* mRNA levels and the diminished DILP2 staining of *dWFS1* and *dCISD* knockout flies (Fig. 1H–J).

These results indicate that restoring the decreased activity of IP$_3$R in *dWFS1* and *dCISD* mutants can effectively ameliorate their diabetes-like phenotypes.

## WFS1 and CISD2 interact with IP$_3$R in an independent manner

Having shown that both WFS1 and CISD2 regulate IP$_3$R activity, we sought to reveal the underlying mechanisms. Previous studies have reported that IP$_3$R is regulated by numerous intricate mechanisms—such as intracellular calcium concentration, organelle crosstalk, and upstream stimulating factors—among which protein-protein interaction stands as a major mechanism (Bononi et al, 2017; Chen et al, 2004; Ham et al, 2023; Nguyen et al, 2019; Park et al, 2017; Wu and Bowen, 2008). Therefore, we performed co-immunoprecipitation assays (co-IP) to determine whether WFS1 and CISD2 regulate IP$_3$R activity via physical binding. As expected, WFS1 was observed to bind to IP$_3$R1 under overexpression conditions (Fig. 2C). We also conducted additional co-IP assays using endogenous proteins to further validate the interaction between WFS1 and IP$_3$R1 (Fig. EV1). After confirming that WFS1 binds to IP$_3$R1, we sought to identify the binding region of WFS1 by conducting co-IP using two truncated forms of WFS1 protein (Fig. 2A). The mutant of which luminal C-terminus was eliminated (ΔC) was observed to interact with IP$_3$R1 alongside WT WFS1 (Fig. 2C). However, the mutant with cytosolic N-terminal deletion (ΔN) failed to interact with IP$_3$R1, demonstrating that the N-terminal of WFS1 is key for IP$_3$R1 binding (Fig. 2C). We also conducted co-IP of IP$_3$R1, WT CISD2, and a C-terminally truncated CISD2 (D1) (Fig. 2B). Whereas WT CISD2 physically interacted with IP$_3$R1, the D1 form did not (Fig. 2D).

We then further sought to narrow down which amino acid residue within the cytosolic C-terminus of CISD2 is crucial for to the protein-protein interaction of CISD2 and IP$_3$R1. By aligning the protein sequences of CISD2 across different species, we found that CISD2 contains Fe-S cluster binding sites within the CDGSH domain at the cytosolic C-terminus that are highly conserved across various species (Appendix Fig. S6A). To identify the binding site of CISD2 that is responsible for its interaction with IP$_3$R1, we focused on the conserved residues in the Fe-S cluster binding sites, such as cysteine 99, cysteine 101, cysteine 110, and histidine 114. We generated point mutants of CISD2 in which each of these residues was substituted with alanine, and assessed their binding affinity to IP$_3$R1. While CISD2 C101A mutant lost its binding ability towards IP$_3$R1, CISD2 C99A, CISD2 C110A, and CISD2 H114A retained their binding abilities (Fig. 2E; Appendix Fig. S6B). This result suggests that cysteine 101 of CISD2 confers its binding ability to IP$_3$R and may play a critical role in the regulation of IP$_3$R activity.

Next, having demonstrated that both WFS1 and CISD2 physically interact with IP$_3$R, we attempted to determine whether these two proteins are dependent on each other in protein-protein interaction with IP$_3$R. Even when CISD2 was overexpressed dose-dependently, there was no difference of the interaction between WFS1 and IP$_3$R1 (Fig. 2F). Similarly, under dose-dependent overexpression of WFS1, the interaction between CISD2 and IP$_3$R1 did not change (Fig. 2G). Also, the interaction between WFS1 and IP$_3$R1 was invariable regardless of whether CISD2 was absent or overexpressed (Fig. 2H). Likewise, CISD2 interacted with IP$_3$R1 even in the absence or overexpression of WFS1 (Fig. 2I).

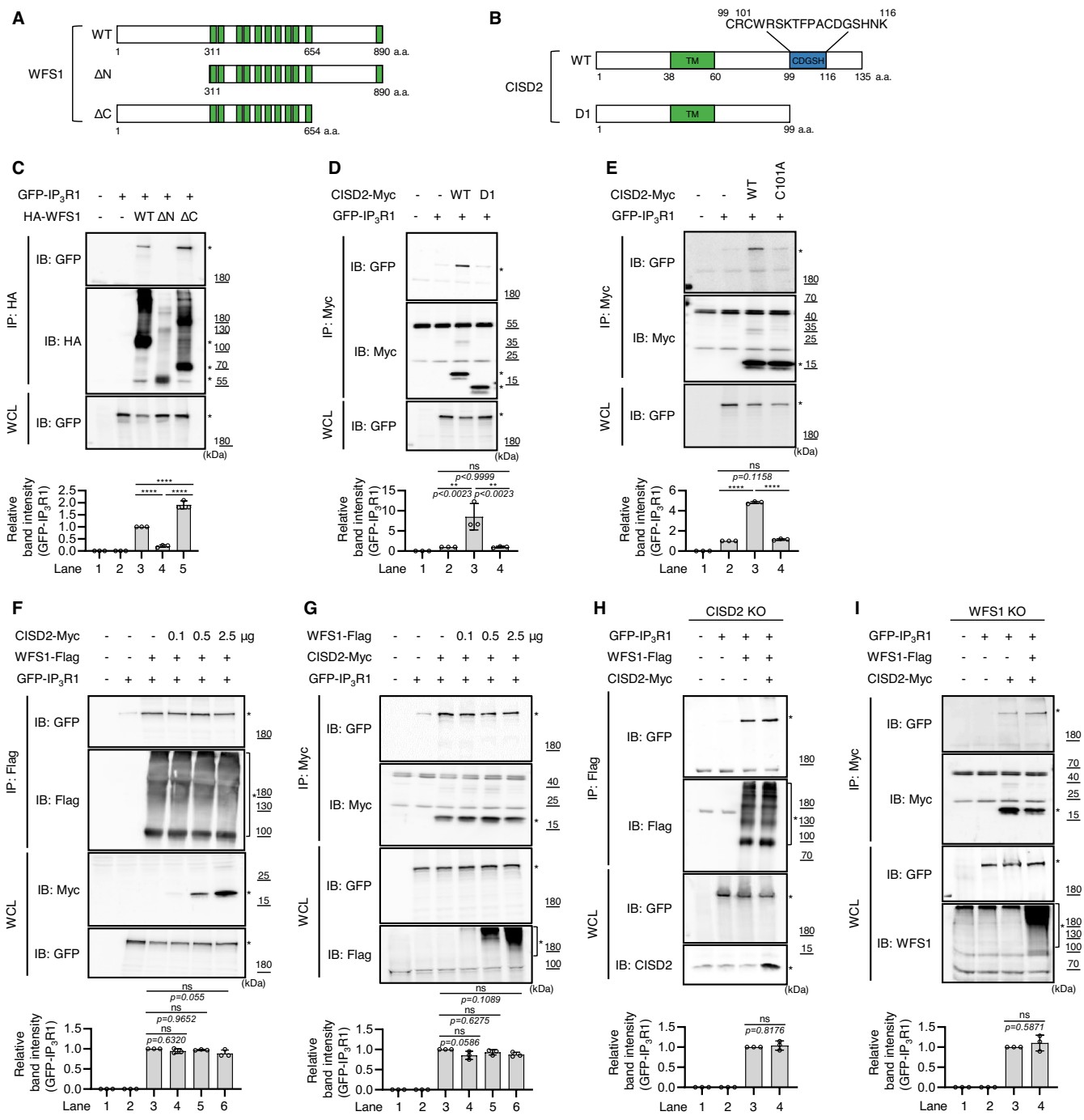

**Figure 2. WFS1 and CISD2 interact with IP$_3$R independently.**

(A) Schematic diagram representing truncated forms of WFS1. The transmembrane domains (TM) were designated with green boxes. (B) Schematic diagram representing various forms of CISD2. The TM and CDGSH domain were designated. (C) HEK293 cells were transfected as indicated and cell lysates were subjected to anti-HA immunoprecipitation followed by immunoblot analysis. (D, E, G) HEK293 cells were transfected as indicated and cell lysates were subjected to anti-Myc immunoprecipitation followed by immunoblot analysis. (F) HEK293 cells were transfected as indicated and cell lysates were subjected to anti-Flag immunoprecipitation followed by immunoblot analysis. (H) CISD2 KO HEK293 cells were transfected as indicated and cell lysates were subjected to anti-Flag immunoprecipitation followed by immunoblot analysis. (I) WFS1 KO HEK293 cells were transfected as indicated and cell lysates were subjected to anti-Myc immunoprecipitation followed by immunoblot analysis. The asterisks denote the band of interest. Bottom bar graphs show relative quantification of the band intensity for the interaction between WFS1 (C, F, H) or CISD2 (D, E, G, I) and IP$_3$R1 shown by co-immunoprecipitation experiments ($n = 3$). Data information: All figures are representatives of three independent experiments. All quantifications were analyzed by one-way ANOVA with Tukey multiple-comparison test. **$p < 0.01$, ****$p < 0.0001$. ns, not significant. All data are presented as mean ± SD. Source data are available online for this figure.

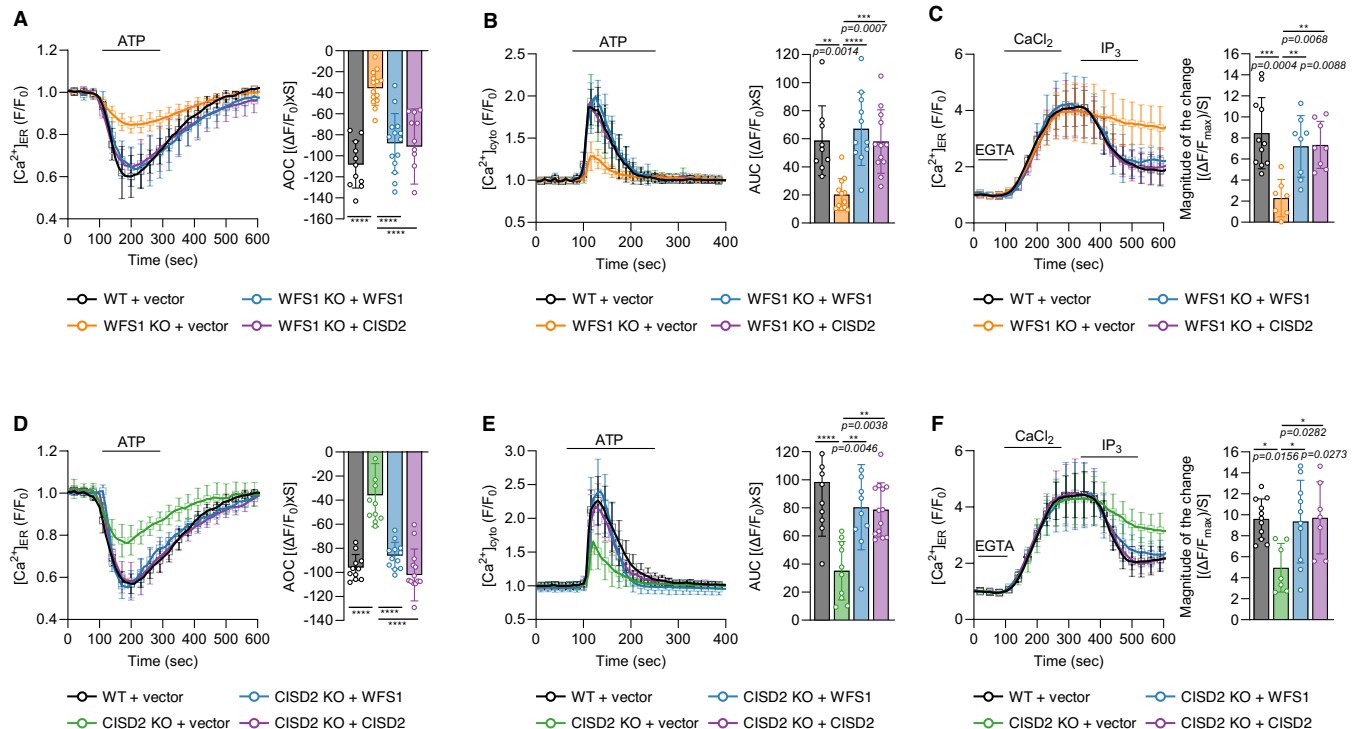

**Figure 3. Disrupted ER calcium efflux induced by loss of WFS1 or CISD2 is mutually rescued by the overexpression of either gene.**

(A) Measurement of ER calcium modulation in WT HEK293 cells transfected with empty vector (black, $n = 95$ cells, 11 coverslips) and WFS1 KO HEK293 cells transfected with empty vector (orange, $n = 71$ cells, 14 coverslips), *WFS1* (blue, $n = 58$ cells, 14 coverslips), or *CISD2* (purple, $n = 61$ cells, 11 coverslips). (B) Measurement of cytosolic calcium modulation in WT HEK293 cells transfected with empty vector (black, $n = 102$ cells, 9 coverslips) and WFS1 KO HEK293 cells transfected with empty vector (orange, $n = 98$ cells, 12 coverslips), *WFS1* (blue, $n = 77$ cells, 11 coverslips), or *CISD2* (purple, $n = 108$ cells, 12 coverslips). The right-side bar graphs indicate the quantification of the normalized calcium traces using AOC or AUC of calcium fluxes during ATP treatment. (C) Measurement of the IP₃R activity of WT HEK293 cells transfected with empty vector (black, $n = 79$ cells, 11 coverslips) and WFS1 KO HEK293 cells transfected with empty vector (orange, $n = 54$ cells, 7 coverslips), *WFS1* (blue, $n = 53$ cells, 8 coverslips), or *CISD2* (purple, $n = 53$ cells, 8 coverslips). The right-side bar graph represents the magnitude of the change during IP₃ treatment. (D) Identical experiments measuring ER calcium modulation were conducted in WT HEK293 cells transfected with empty vector (black, $n = 103$ cells, 11 coverslips) and CISD2 KO HEK293 cells transfected with empty vector (green, $n = 104$ cells, 11 coverslips), *WFS1* (blue, $n = 84$ cells, 12 coverslips), or *CISD2* (purple, $n = 91$ cells, 14 coverslips). (E) Identical experiments measuring cytosolic calcium modulation were conducted in WT HEK293 cells transfected with empty vector (black, $n = 124$ cells, 11 coverslips) and CISD2 KO HEK293 cells transfected with empty vector (green, $n = 118$ cells, 10 coverslips), *WFS1* (blue, $n = 119$ cells, 10 coverslips), or *CISD2* (purple, $n = 123$ cells, 13 coverslips). The right-side bar graphs indicate the quantification of the normalized calcium traces using AOC or AUC of calcium fluxes during ATP treatment. (F) Measurement of the IP₃R activity of WT HEK293 cells transfected with empty vector (black, $n = 70$ cells, 11 coverslips) and CISD2 KO HEK293 cells transfected with empty vector (green, $n = 73$ cells, 7 coverslips), *WFS1* (blue, $n = 55$ cells, 10 coverslips), or *CISD2* (purple, $n = 77$ cells, 7 coverslips). The right-side bar graph represents the magnitude of the change during IP₃ treatment. Data information: All figures are representatives of three or more independent experiments. All quantifications were analyzed by one-way ANOVA with Tukey multiple-comparison test. $*p < 0.05$, $**p < 0.01$, $***p < 0.001$, $****p < 0.0001$. All data are presented as mean ± SD. Source data are available online for this figure.

Collectively, these results suggest that WFS1 and CISD2 bind to IP₃R in an independent manner.

## Disrupted ER calcium efflux induced by loss of WFS1 or CISD2 is reciprocally rescued by WFS1 or CISD2 overexpression

Given that WFS1 and CISD2 independently bind to IP₃R, we investigated if these two proteins could reciprocally rescue the disrupted intracellular calcium fluxes induced by the deficiency of either gene. The decreased ER calcium release, cytosolic calcium levels, and IP₃R activity in WFS1 knockout cells were rescued by either WFS1 or CISD2 overexpression (Fig. 3A–C). Correspondingly, overexpression of WFS1 or CISD2 also redeemed the decreased ER calcium release, cytosolic calcium levels, and IP₃R activity in CISD2 knockout cells (Fig. 3D–F). Moreover, to determine whether overexpression of WFS1 or CISD2

affects intracellular calcium flux in the presence of endogenous WFS1 and CISD2, we overexpressed WFS1 or CISD2 in WT HEK293 cells and measured calcium fluxes and IP₃R activity. We found that overexpression of WFS1 or CISD2 augments ER calcium release, cytosolic calcium levels, and IP₃R activity in the presence of endogenous WFS1 and CISD2 (Fig. EV2A–C). Overall, these data support that WFS1 and CISD2 increase IP₃R calcium channel activity independently.

## Diabetes-like phenotypes in *dWFS1* mutant and *dCISD* mutant fly lines are alleviated by the overexpression of either gene

As our mammalian cell experiments indicate that WFS1 and CISD2 independently elevate IP₃R activity, we sought to validate the complementary effects of these two proteins in vivo. Prior to examining the physiological effect of overexpressing *dWFS1* and

*dCISD* in flies, we verified protein levels of dWFS1 and dCISD in *tub>LacZ*, *tub>dWFS1*, and *tub>dCISD* transgenic flies using immunoblot analysis. The relative protein levels of dWFS1 and dCISD in transgenic flies were determined to be approximately 7-fold and 3-fold higher, respectively, compared to the endogenous protein level (Appendix Fig. S7A,B). We employed the transgenic flies, of which protein levels were confirmed, to test whether overexpression of *dWFS1* and *dCISD* could complement the absence of one another to ameliorate diabetes-like phenotypes in mutant flies. As expected, ubiquitous overexpression of *dWFS1* using the *tub-Gal4* driver in *dWFS1* mutant flies rescued elevated hemolymph glucose and TAG levels (Fig. 4A,B). Additionally, *dCISD* overexpression alleviated these phenotypes of *dWFS1* mutant flies (Fig. 4A,B). Overexpression of either *dWFS1* or *dCISD* also ameliorated the abnormal GTT response of *dWFS1* mutant flies, diminishing the jump in hemolymph glucose after feeding and quickening the recovery during the subsequent fasting (Fig. 4C,D). Likewise, the increased hemolymph glucose and TAG levels of *dCISD* knockout flies were rescued by the ubiquitous overexpression of either *dWFS1* or *dCISD* (Fig. 4E,F). The abnormal GTT response of *dCISD* mutant flies were also restored by the overexpression of either gene (Fig. 4G,H). In addition, the lower *dilp2* mRNA levels of *dWFS1* and *dCISD* mutant flies were ameliorated by the expression of either *dWFS1* or *dCISD* (Fig. 4I,J). Correspondingly, diminished DILP2 staining levels of *dWFS1* and *dCISD* mutants were rescued by the overexpression of either gene (Fig. 4K,L). These in vivo results indicate that dWFS1 and dCISD can complement each other, filling in for the absence of one another to alleviate diabetes-like phenotypes. Thus, our results suggest that the overexpression of either gene is sufficient to not only restore the decreased IP$_3$R activity caused by deficits in WFS1 or CISD2 but also ameliorate the diabetic complications that WS patients experience.

## Overexpression of CISD2 peptide increases IP$_3$R activity

As we uncovered the binding region of CISD2 with IP$_3$R above, we postulated that a peptide consisting of this binding sequence can directly bind to IP$_3$R and sufficiently control its activity. We generated N-terminal Flag-tagged vectors that express peptides of several lengths, containing amino acid sequences from the C-terminus of CISD2, crucial for CISD2-IP$_3$R binding as shown above. The peptides we generated have lengths of 18, 16, and 14 amino acids, and consist of the following sequences: N'-CRCWRSKTFPACDGSHNK-C', N'-CRCWRSKTFPACDGSH-C', and N'-CWRSKTFPACDGSH-C', respectively (Figs. 5A and EV3A). Next, we performed dot blot assays to verify the protein expression levels of each CISD2 peptide. We transfected HEK293T cells with each CISD2 peptide, and the cell lysates were subjected to immunoprecipitation (IP) using anti-Flag antibodies. The Flag-tagged CISD2 peptides were then detected with anti-Flag antibodies. We confirmed that the peptides with different lengths were expressed at similar levels (Fig. EV3B) and proceeded with subsequent calcium imaging experiments. Overexpression of the peptides of 16 and 14 amino acids in length did not significantly rescue the decreased ER calcium release of WFS1 knockout cells (Fig. EV3C). However, we observed that our peptide consisting of 18 amino acids fully rescued the impaired ER calcium release of WFS1-deficient cells (Fig. EV3C). We therefore conducted subsequent experiments with our CISD2 peptide of 18 amino acids in length, hereinafter referred to as WT CISD2 peptide.

To determine if this WT CISD2 peptide binds to IP$_3$R directly, we conducted co-IP assays with WT CISD2 peptide, C101A CISD2 peptide and IP$_3$R1 in WT and CISD2 KO HEK293 cells. We found that WT CISD2 peptide interacted with IP$_3$R1 in WT HEK293 cells whereas the C101A CISD2 peptide did not (Fig. EV4A). In the absence of endogenous CISD2, WT CISD2 peptide retained its binding ability to IP$_3$R1 while C101A CISD2 peptide remained unbound (Fig. EV4A). Additionally, we conducted co-IP between CISD2 and IP$_3$R1 under dose-dependent overexpression conditions of WT CISD2 peptide. Increased expression of WT CISD2 peptide subsequently decreased the degree of binding between CISD2 and IP$_3$R1 (Fig. 5B, left panels). However, dose-dependent overexpression of the C101A CISD2 peptide, consisting of the binding region of CISD2 with the key cysteine for IP$_3$R1 binding replaced by alanine, did not impact the interaction between CISD2 and IP$_3$R1 (Fig. 5B, right panels). These results suggest that WT CISD2 peptide directly binds to IP$_3$R and occupies the same binding region as CISD2.

Moreover, overexpression of WT CISD2 peptide rescued the decreased ER calcium release, cytosolic calcium levels, and IP$_3$R activity in WFS1 (Fig. 5C–E) or CISD2 knockout cells (Fig. 5F–H). When WT CISD2 peptide was overexpressed in WT cells, ER calcium release, cytosolic calcium levels, and IP$_3$R activity were increased compared to the WT cells expressing an empty vector (Fig. EV2A–C). Moreover, we compared WT CISD2 peptide's effect on ER calcium release in WT and CISD2 knockout cells in the same experiment. When WT CISD2 peptide was overexpressed, ER calcium release increased by approximately 40 units both in WT and CISD2 knockout cells (Fig. EV2D). Also, cytosolic calcium levels were elevated by similar levels in WT and CISD2 lacking cells in the same experiment (Fig. EV2E). We also verified that overexpression of C101A CISD2 peptide failed to rescue the decreased ER calcium release and cytosolic calcium levels in WFS1 knockout cells in contrast to WT CISD2 peptide (Fig. EV4B,C). Comprehensively, our results support that WT CISD2 peptide directly binds to IP$_3$R and controls its calcium channel activity.

We also tested whether the exogenously synthesized WT CISD2 peptide could restore disrupted calcium fluxes caused by the loss of WFS1 or CISD2, when delivered into cells. To deliver WT CISD2 peptide into the cells, we used penetratin, a cell-penetrating peptide (CPP). We put the penetratin sequence at the C-terminus of WT CISD2 peptide and GGGS linker in between WT CISD2 peptide and CPP as a spacer. Rhodamine B was conjugated to CPP as a fluorescence tag (Fig. EV5A). We also synthesized CPP without CISD2 peptide sequence, hereinafter referred as CPP, to use as control (Fig. EV5A). We incubated HEK293 cells with buffer alone, CPP, or WT CISD2-CPP, and observed that CPP and WT CISD2-CPP resided within the cells (Fig. EV5B), thus verifying that our WT CISD2-CPP effectively penetrated the cell membrane. We also measured ER calcium release under WT CISD2-CPP incubation. WT CISD2-CPP treatment redeemed the reduced ER calcium flux of WFS1 or CISD2 knockout cells, but treatment of just CPP alone were comparable to the buffer-treated control cells (Fig. EV5C,D). Also, incubation with CPP did not change the ER calcium release in HEK293 WT cells, but incubation with WT CISD2-CPP augmented ER calcium release compared to the buffer-incubated cells (Fig. EV5C,D). These results suggest that delivery of WT CISD2 peptide through the cell membrane rescues impaired ER calcium flux induced by WFS1 or CISD2 deficiency.

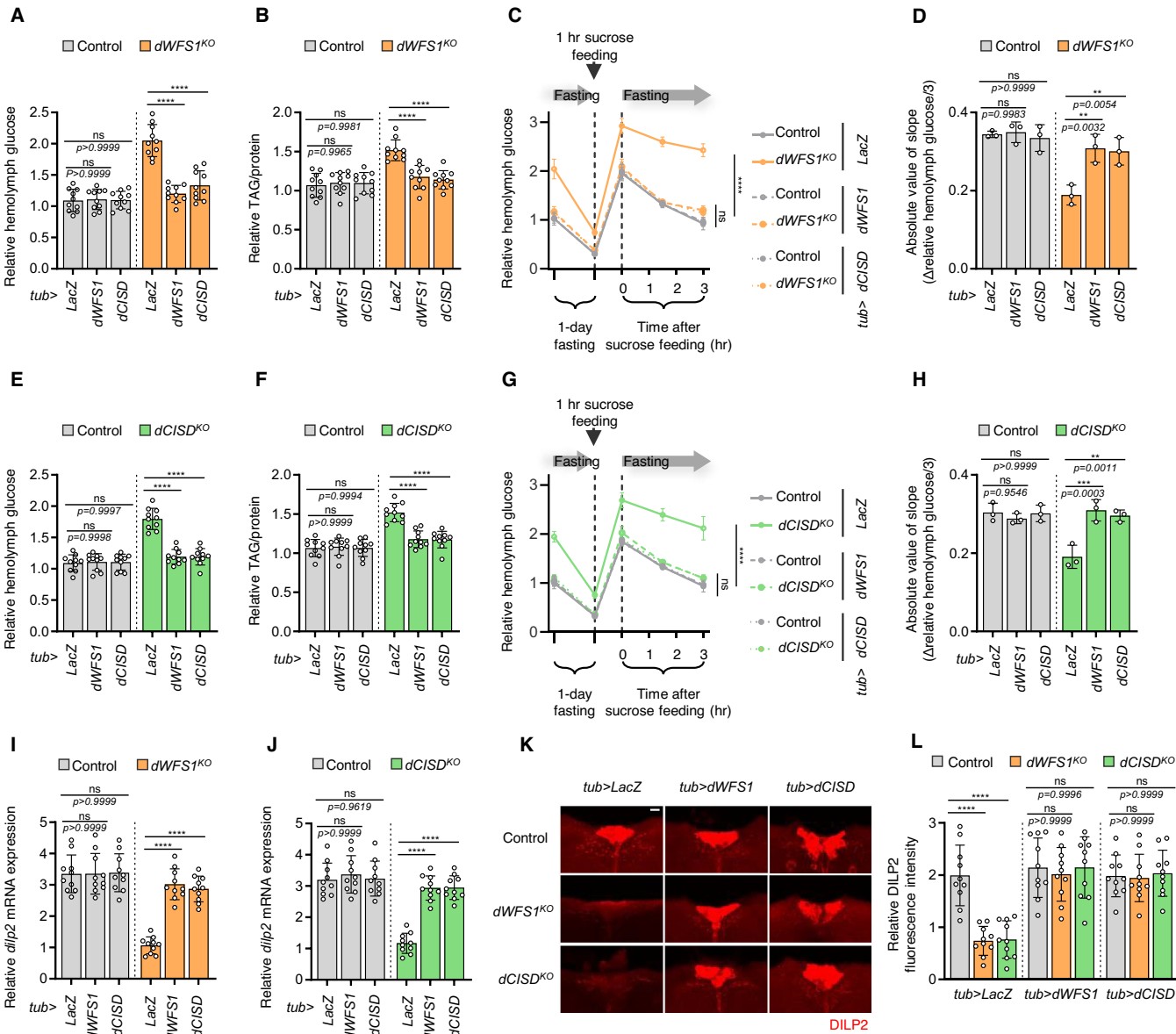

**Figure 4. Overexpression of *dWFS1* or *dCISD* reciprocally mitigates diabetes-like phenotypes in *dWFS1* and *dCISD* mutant flies.**

(A) Relative levels of hemolymph glucose from 30-day-old *w1118* or *dWFS1* KO flies expressing UAS-*LacZ*, UAS-*dWFS1*, or UAS-*dCISD* with the *tub-Gal4* driver, normalized to *tub>LacZ* fly controls (*n* = 10). (B) Relative levels of TAG divided by their respective protein levels of 30-day-old *w1118* or *dWFS1* KO flies expressing *LacZ*, *dWFS1*, or *dCISD*, normalized to *tub>LacZ* flies (*n* = 10). (C) Relative levels of hemolymph glucose of 30-day-old *w1118* or *dWFS1* KO flies expressing *LacZ*, *dWFS1*, or *dCISD* followed through the glucose tolerance test, normalized to *tub>LacZ* flies (*n* = 3). ns indicates *p*-values of 0.0757 (control, *tub>LacZ* vs. *dWFS1* KO, *tub>dWFS1*) and 0.2186 (control, *tub>LacZ* vs. *dWFS1* KO, *tub>dCISD*). (D) The absolute value of the slope after the 1 h sucrose feeding to quantify the recovery of heightened glucose levels (*n* = 3). (E) Relative levels of hemolymph glucose from 30-day-old *w1118* or *dCISD* KO flies expressing *LacZ*, *dWFS1*, or *dCISD*, normalized to *tub>LacZ* flies (*n* = 10). (F) Relative levels of TAG divided by their respective protein levels of 30-day-old *w1118* or *dCISD* KO flies expressing *LacZ*, *dWFS1*, or *dCISD*, normalized to *tub>LacZ* flies (*n* = 10). (G) Relative levels of hemolymph glucose of 30-day-old *w1118* or *dCISD* KO flies expressing *LacZ*, *dWFS1*, or *dCISD* followed through the glucose tolerance test, normalized to *tub>LacZ* flies (*n* = 3). ns indicates *p*-values of 0.5779 (control, *tub>LacZ* vs. *dCISD* KO, *tub > dWFS1*) and 0.5509 (control, *tub>LacZ* vs. *dCISD* KO, *tub>dCISD*). (H) The absolute value of the slope after the 1 h sucrose feeding to quantify the recovery of heightened glucose levels (*n* = 3). (I) Relative *dilp2* mRNA levels in whole bodies of 30-day-old *w1118* and *dWFS1* KO flies expressing *LacZ*, *dWFS1*, or *dCISD*, as quantified through real-time PCR, normalized to *dWFS1* KO, *tub>LacZ* flies (*n* = 10). (J) Relative *dilp2* mRNA levels in *w1118* and *dCISD* KO flies expressing *LacZ*, *dWFS1*, or *dCISD*, normalized to *dCISD* KO, *tub>LacZ* flies (*n* = 10). (K) Staining of fly brains using anti-DILP2 antibody (*n* = 10). Scale bar, 10 μm. (L) Quantifications of the fluorescence intensity of DILP2 brain images, normalized to *dWFS1* KO, *tub>LacZ* flies (*n* = 10). Data information: All figures are representatives of three or more independent experiments. All quantifications were analyzed by one-way ANOVA with Tukey multiple-comparison test. \*\**p* < 0.01, \*\*\**p* < 0.001, \*\*\*\**p* < 0.0001. ns, not significant. All data are presented as mean ± SD. Source data are available online for this figure.

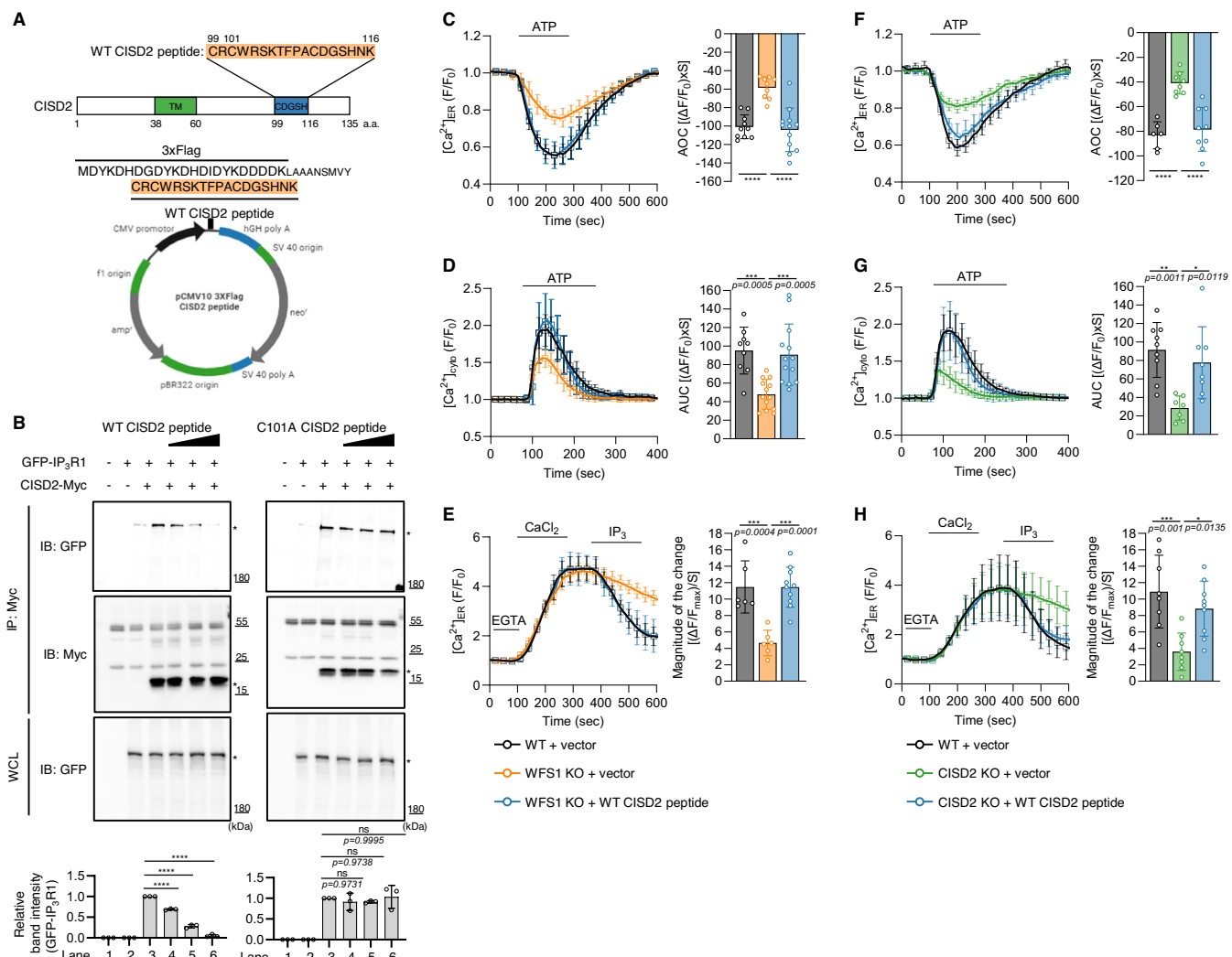

**Figure 5. Overexpression of CISD2 peptide recovers diminished IP₃R activity in WFS1 or CISD2 lacking cells via interacting with IP₃R.**

(A) Schematic representation of vector map harboring WT CISD2 peptide sequence, highlighted in orange. (B) HEK293T cells were transfected as indicated and cell lysates were subjected to anti-Myc immunoprecipitation followed by immunoblot analysis. The asterisks denote the band of interest. Bottom bar graphs show relative quantification of the band intensity of anti-GFP blot normalized to anti-Myc blot of immunoprecipitated proteins by anti-Myc antibody ($n = 3$). (C) Measurement of ER calcium modulation in WT HEK293 cells transfected with empty vector (black, $n = 99$ cells, 10 coverslips) and WFS1 KO HEK293 cells transfected with empty vector (orange, $n = 95$ cells, 9 coverslips) or *WT CISD2 peptide* (blue, $n = 84$ cells, 12 coverslips). (D) Measurement of cytosolic calcium modulation in WT HEK293 cells transfected with empty vector (black, $n = 158$ cells, 9 coverslips) and WFS1 KO HEK293 cells transfected with empty vector (orange, $n = 124$ cells, 13 coverslips) or *WT CISD2 peptide* (blue, $n = 127$ cells, 13 coverslips). The right-side bar graphs indicate the quantification of the normalized calcium traces using AOC or AUC of calcium fluxes during ATP treatment. (E) Measurement of the IP₃R activity of WT HEK293 cells transfected with empty vector (black, $n = 76$ cells, 6 coverslips) and WFS1 KO HEK293 cells transfected with empty vector (orange, $n = 73$ cells, 6 coverslips) or *WT CISD2 peptide* (blue, $n = 54$ cells, 10 coverslips). The right-side bar graph represents the magnitude of the change during IP₃ treatment. (F) Identical experiments measuring ER calcium modulation were conducted in WT HEK293 cells transfected with empty vector (black, $n = 62$ cells, 6 coverslips) and CISD2 KO HEK293 cells transfected with empty vector (green, $n = 52$ cells, 8 coverslips) or *WT CISD2 peptide* (blue, $n = 52$ cells, 9 coverslips). (G) Identical experiments measuring cytosolic calcium modulation were conducted in WT HEK293 cells transfected with empty vector (black, $n = 130$ cells, 9 coverslips) and CISD2 KO HEK293 cells transfected with empty vector (green, $n = 125$ cells, 7 coverslips) or *WT CISD2 peptide* (blue, $n = 132$ cells, 8 coverslips). The right-side bar graphs indicate the quantification of the normalized calcium traces using AOC or AUC of calcium fluxes during ATP treatment. (H) Measurement of the IP₃R activity of WT HEK293 cells transfected with empty vector (black, $n = 92$ cells, 8 coverslips) and CISD2 KO HEK293 cells transfected with empty vector (green, $n = 67$ cells, 8 coverslips) or *WT CISD2 peptide* (blue, $n = 57$ cells, 9 coverslips). The right-side bar graph represents the magnitude of the change during IP₃ treatment. Data information: All figures are representatives of three or more independent experiments. All quantifications were analyzed by one-way ANOVA with Tukey multiple-comparison test. *$p < 0.05$, **$p < 0.01$, ***$p < 0.001$, ****$p < 0.0001$. ns, not significant. All data are presented as mean ± SD. Source data are available online for this figure.

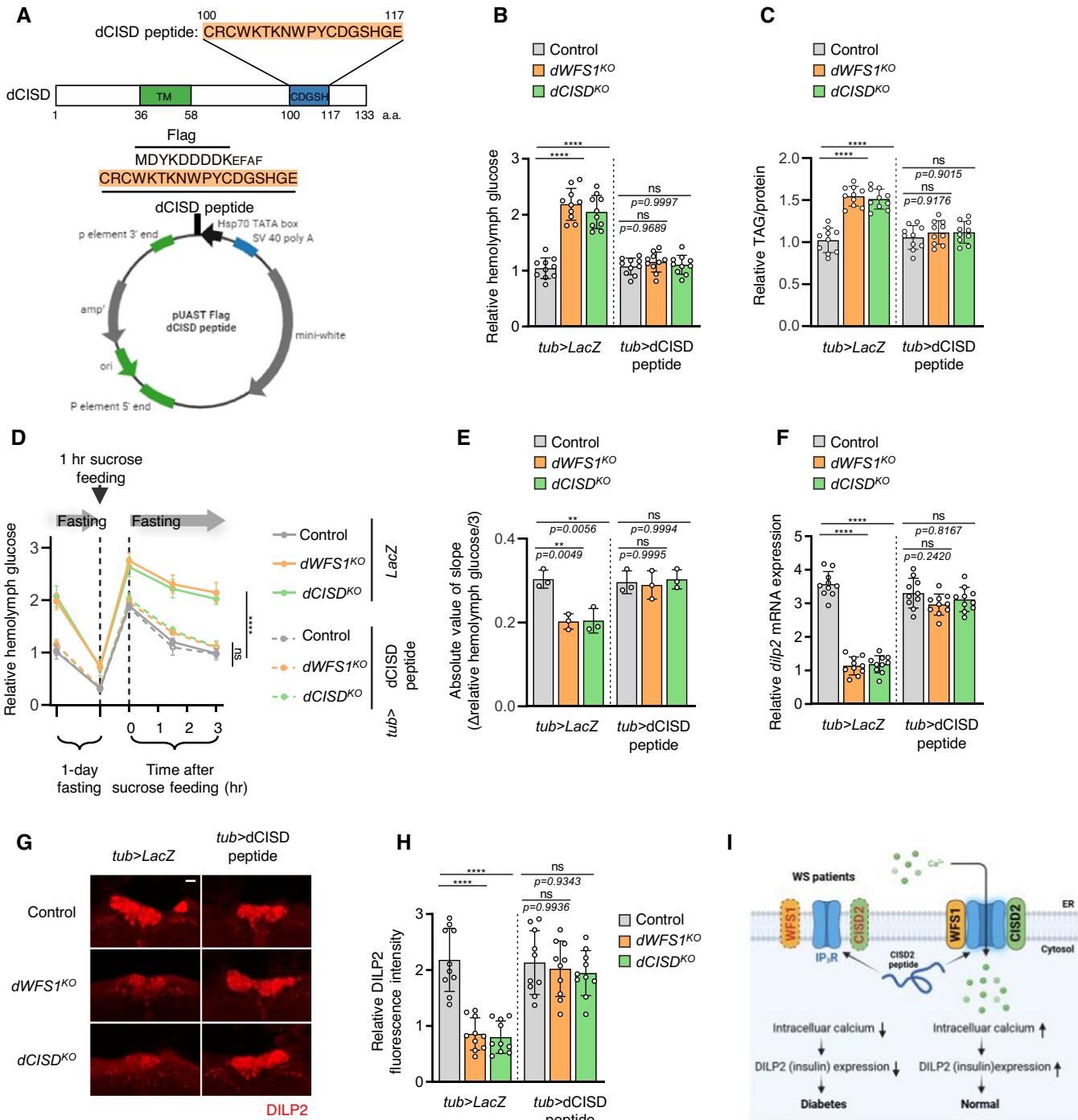

## Overexpression of *Drosophila* dCISD peptide ameliorates diabetes-like phenotypes of *dWFS1* mutant and *dCISD* mutant flies

To confirm that our WT CISD2 peptide can rescue diabetes-like phenotypes of *dWFS1* and *dCISD* mutant flies, we generated transgenic flies expressing the analogous dCISD peptide. We cloned the DNA sequence that codes for the dCISD peptide, N'-CRCWKTKNWPYCDGSHGE-C', which is homologous to the

human WT CISD2 peptide, into the pUAST vector (Fig. 6A). Subsequently, we injected this plasmid into fly embryos and sorted flies effectively expressing our construct by crossing flies with the *tub-Gal4* driver and examining mRNA levels and protein levels through qPCR analysis and dot blot assays, respectively (Appendix Fig. S8A,B). We thus confirmed and established the stable UAS-dCISD peptide line utilized in our experiments. Consistent with our mammalian cell results, ubiquitous overexpression of the dCISD peptide using the *tub-Gal4* driver ameliorated the elevated

◄

**Figure 6. dCISD2 peptide overexpression ameliorates diabetes-like phenotypes of *dWFS1* and *dCISD* mutant flies.**

(A) Scheme of the pUAST vector with the DNA insertion coding for the dCISD peptide, highlighted in orange. (B) Relative levels of hemolymph glucose from 30-day-old *w1118*, *dWFS1* KO, or *dCISD* KO flies expressing either UAS-LacZ or UAS-dCISD peptide with the *tub-Gal4* driver. Data were normalized to *tub>LacZ* controls ($n = 10$). (C) Relative levels of TAG divided by their respective protein levels of 30-day-old flies, normalized to *tub>LacZ* flies ($n = 10$). (D) Relative levels of hemolymph glucose of 30-day-old flies followed through the glucose tolerance test, normalized to *tub>LacZ* flies ($n = 3$). ns indicates *p*-values of 0.8848 (control, *tub>LacZ* vs. *dWFS1* KO, *tub>*dCISD peptide) and 0.8090 (control, *tub>LacZ* vs. *dCISD* KO, *tub>*dCISD peptide). (E) The absolute value of the slope after the 1 h sucrose feeding to quantify the recovery of heightened glucose levels ($n = 3$). (F) Relative *dilp2* mRNA levels in whole fly bodies as quantified through real-time PCR, normalized to *tub>LacZ* flies ($n = 10$). (G) Staining of fly brains using anti-DILP2 antibody ($n = 10$). Scale bar, 10 μm. (H) Quantifications of the fluorescence intensity of DILP2 brain images, normalized to *dWFS1* KO, tub>*LacZ* flies ($n = 10$). (I) A diagram of our proposed model of WS pathogenesis. Normally, both WFS1 and CISD2 bind to and activate IP$_3$R to maintain homeostasis of intracellular calcium. WS patients with mutations in either WFS1 or CISD2 exhibit impaired IP$_3$R activity that can be restored by overexpressing either gene or providing supplements of an 18 amino acid long CISD2 peptide. Data information: All figures are representatives of three or more independent experiments. All quantifications were analyzed by one-way ANOVA with Tukey multiple-comparison test. **$p < 0.01$, ****$p < 0.0001$. ns, not significant. All data are presented as mean ± SD. Source data are available online for this figure.

hemolymph glucose and TAG levels of both *dWFS1* and *dCISD* mutant flies (Fig. 6B,C). In addition, overexpression of the dCISD peptide rescued the abnormal GTT responses shown by *dWFS1* and *dCISD* mutant flies to exhibit hemolymph glucose recovery phenotypes similar to those of control flies, restoring the delayed return of elevated hemolymph glucose levels (Fig. 6D,E). Overexpression of the dCISD peptide also rescued the diminished *dilp2* mRNA levels as well as anti-DILP2 staining of *dWFS1* and *dCISD* mutant flies (Fig. 6F–H). These results demonstrate that overexpression of the dCISD peptide is effective and sufficient for ameliorating the diabetes-like phenotypes observed in our WS model flies.

Overall, our results propose a model in which mutations in *WFS1* or *CISD2* reduce IP$_3$R activity to impair ER calcium release and induce diabetic complications. In turn, overexpression of either WS gene rescues IP$_3$R activity and ameliorates diabetic phenotypes. Furthermore, overexpression of CISD2 peptide restores IP$_3$R activity and reinstates calcium homeostasis to alleviate WS-related diabetic complications induced by deficits in *WFS1* or *CISD2* (Fig. 6I).

## Discussion

In this study, we established that knockout of WS causative genes, *WFS1* and *CISD2*, in mammalian cells decreases IP$_3$R activity. Our WS *Drosophila* models exhibited diabetes-like phenotypes that were normalized by increasing IP$_3$R activity. Additionally, we demonstrated that both WFS1 and CISD2 directly bind to IP$_3$R, activating its function. The decreased ER calcium release in WFS1 and CISD2 knockout cells was rescued by overexpression of either WFS1 or CISD2, which reinstated IP$_3$R activity. Notably, we identified a CISD2 peptide that alleviates ER calcium defects of WFS1- or CISD2-deficient cells by solely increasing IP$_3$R activity, further showing that this peptide mitigates diabetes-like phenotypes as well. With these results, we propose our CISD2 peptide as a new therapeutic tool that successfully restores IP$_3$R activity to effectively ameliorate the diabetic complications associated with WS.

Amongst the various proteins known to manipulate IP$_3$R activity (Bononi et al, 2017; Chen et al, 2004; Ham et al, 2023; Nguyen et al, 2019; Park et al, 2017; Wu and Bowen, 2008), WFS1 and CISD2 have been further established as IP$_3$R regulators in this study. In previous studies, another ER membrane protein, BCL2 has also been known to interact with CISD2 (Chang et al, 2012a;

Chang et al, 2010; Chang et al, 2012b). However, we observed that CISD2 maintained its binding affinity for IP$_3$R1 even under BCL2 knockdown conditions (Appendix Fig. S9). This finding supports that CISD2 directly interacts with IP$_3$R without the involvement of another binding protein, BCL2. Moreover, our examination of the interaction between WFS1 and CISD2 with truncated forms of IP$_3$R1, consisting of the cytosol-exposed N-terminus domains (SD, LBD, MD1, MD2, and MD3) and the cytosol-exposed C-terminus GK domain, revealed a common binding of WFS1 and CISD2 with IP$_3$R1's LBD, MD1, MD2, and GK domains (Appendix Fig. S10A–C). However, we note that the SD domain of IP$_3$R1 interacts with WFS1 but not with CISD2 (Appendix Fig. S10B,C). These experiments suggest that WFS1 and CISD2 interact independently at distinct sites on IP$_3$R to manipulate its activity, demonstrating their roles as IP$_3$R regulators and emphasizing ER calcium dysfunction as a major mechanism of WS-related diabetes. As diabetes is just one of the numerous symptoms presented in WS, it would be of interest to study whether reinstatement of IP$_3$R activity could rescue other manifestations including optic nerve atrophy, deafness, urinary tract defects, and neurological impairments. IP$_3$R defects have been connected to neurodegenerative diseases like Alzheimer's and Huntington's, highlighting the pivotal role of IP$_3$R in various pathologies and the necessity to understand its regulation.

In addition, previous studies have illustrated the role of calcium in WS-independent diabetes (Eshima, 2021; Eshima et al, 2014; Madec et al, 2021). Reductions in ER calcium of β-cells have been related to both type 1 and type 2 diabetes, and such calcium dysregulation leads to defects in insulin secretion or synthesis of β-cell (Kim et al, 2025; Klec et al, 2019; Sabatini et al, 2019; Zhang et al, 2020). Dysregulation of ER calcium leakage through ryanodine receptors (RyR) and impaired ER calcium uptake through SERCA have also been noted in diabetic β-cells (Liang et al, 2014; Sabatini et al, 2019; Yamamoto et al, 2019). Store-operated calcium entry, another route through which ER calcium is replenished, is impaired in diabetic conditions as well (Kono et al, 2018). IP$_3$R also transfers ER calcium to mitochondria through MAMs, and the subsequent mitochondrial calcium accumulation is crucial for numerous physiological processes (Atakpa-Adaji and Ivanova, 2023; Hamada and Mikoshiba, 2020; Madec et al, 2021; Proulx et al, 2021). Pyruvate dehydrogenase phosphatase 1, functioning in the Krebs cycle, regulates pyruvate dehydrogenase (PDH) activity and determines the metabolic fate of pyruvate. Low mitochondrial calcium uptake may result in impaired PDH activity, a phenotype observed in the rodent model of chronic type 1 diabetes

mellitus (Cividini et al, 2021; Elnwasany et al, 2023; Glancy and Balaban, 2012; Park et al, 2018; Patel and Korotchkina, 2006; Tabatabaei Dakhili et al, 2023). Activation of PDH has thus been proposed as a therapy of interest for metabolic diseases including diabetes (Jeon et al, 2021; Jiang et al, 2019; Le Page et al, 2015; Tabatabaei Dakhili et al, 2023). While we did not measure mitochondrial calcium in our experiments, alterations in ER calcium leads to changes in calcium across various organelles including mitochondria, affecting their functions. Ultimately, proper control of intracellular calcium is crucial for preventing diabetes.

Several proposed treatments for WS target intracellular calcium homeostasis (Abreu and Urano, 2019; Mishra et al, 2021; Rigoli et al, 2022). Dantrolene sodium, a derivative of hydantoin and a skeletal muscle relaxant (Ellis et al, 1973), inhibits RyR located on the ER membrane and reduces cytosolic calcium (Abreu et al, 2021; Fruen et al, 1997; Paul-Pletzer et al, 2001; Szentesi et al, 2001). Other investigations identified JTV-519 (also known as K201) as a novel candidate compound that binds to and stabilizes RyR2 in its closed state (Rigoli et al, 2022; Wehrens et al, 2004). Valproate, a widely used anti-convulsant and mood-stabilizing medication, mitigates ER stress-induced apoptosis in neuronal and hepatocellular cells as well as protects pancreatic β-cells from palmitate-induced ER stress and apoptosis (Huang et al, 2014; Kakiuchi et al, 2009; Kim et al, 2005; Li et al, 2017). Though current WS treatments seek to stabilize ER calcium by targeting ER stress and RyR, our findings indicate that the underlying cause for reduced cytosolic calcium levels in WS is impaired IP$_3$R activity. Therefore, the fundamental treatment for WS should target and activate IP$_3$R to normalize cellular calcium homeostasis, pushing us to propose our CISD2 peptide as a new therapeutic approach. Peptide treatments are quickly gaining popularity, with recent advances in production and delivery methods (Baig et al, 2018; Wang et al, 2022). Our CISD2 peptide consists of just 18 amino acids, putting it at approximately 2 kDa and falling safely within the range of 1 to 10 kDa that houses most therapeutic peptides (Diao and Meibohm, 2013). We also confirmed that cell-penetrating CISD2 at a small concentration of 10 μM can regulate IP$_3$R activity in mammalian cells. Altogether, we propose our CISD2 peptide as an innovative method to restore ER calcium homeostasis and ameliorate complications associated with WS.

# Methods

### Reagents and tools table

| Reagent/Resource | Reference or Source | Identifier or Catalog Number |
|---|---|---|
| **Experimental models** | | |
| HEK293 cells | Dr. John Blenis (Cornell University, USA) | N/A |
| HEK293T cells | Dr. John Blenis (Cornell University, USA) | N/A |
| WFS1 KO HEK293 cells | This study | N/A |
| CISD2 KO HEK293 cells | This study | N/A |
| *w1118 Drosophila* | Bloomington Drosophila Stock Center | 3605 |
| *tub-Gal4 Drosophila* | Bloomington Drosophila Stock Center | 5138 |
| UAS-*Itpr Drosophila* | Bloomington Drosophila Stock Center | 30742 |
| UAS-*WFS1 Drosophila* | Bloomington Drosophila Stock Center | 8357 |
| UAS-*LacZ Drosophila* | Bloomington Drosophila Stock Center | 1776 |
| *mef2-Gal4 Drosophila* | Bloomington Drosophila Stock Center | 27390 |
| UAS-ERGCaMP6-210 *Drosophila* | Bloomington Drosophila Stock Center | 83294 |
| UAS-GCaMP5G *Drosophila* | Bloomington Drosophila Stock Center | 42037 |
| *WFS1* KO *Drosophila* | This study | N/A |
| UAS-*dCISD Drosophila* | This study | N/A |
| UAS-dCISD peptide *Drosophila* | This study | N/A |
| **Recombinant DNA** | | |
| pcDNA3 WFS1 Flag | Addgene | 13011 |
| pcDNA3 3xHA WFS1 | This study | N/A |
| pcDNA3 3xHA WFS1 ΔN | This study | N/A |
| pcDNA3 3xHA WFS1 ΔC | This study | N/A |
| pcDNA3 CISD2 Myc/His | This study | N/A |
| pcDNA3 CISD2 D1 Myc/His | This study | N/A |
| pcDNA3 CISD2 C99A Myc/His | This study | N/A |
| pcDNA3 CISD2 C101A Myc/His | This study | N/A |
| pcDNA3.1 CISD2 C110A Myc/His | This study | N/A |
| pcDNA3.1 CISD2 H114A Myc/His | This study | N/A |
| pGFPC1 IP$_3$R1 (bovine) | Dr. Sang Ki Park (Postech, Korea) | N/A |
| pcDNA3.1 IP$_3$R1 (bovine) | Dr. Sang Ki Park (Postech, Korea) | N/A |
| pFlag-cmv2 IP$_3$R1 SD (bovine) | Dr. Sang Ki Park (Postech, Korea) | N/A |
| pFlag-cmv2 IP$_3$R1 LBS (bovine) | Dr. Sang Ki Park (Postech, Korea) | N/A |
| pFlag-cmv2 IP$_3$R1 MD1 (bovine) | Dr. Sang Ki Park (Postech, Korea) | N/A |
| pFlag-cmv2 IP$_3$R1 MD2 (bovine) | Dr. Sang Ki Park (Postech, Korea) | N/A |
| pFlag-cmv2 IP$_3$R1 MD3 (bovine) | Dr. Sang Ki Park (Postech, Korea) | N/A |

| Reagent/Resource | Reference or Source | Identifier or Catalog Number |
|---|---|---|
| pFlag-cmv2 IP$_3$R1 GK (bovine) | Dr. Sang Ki Park (Postech, Korea) | N/A |
| pCMV10 3xFlag WT CISD2 peptide | This study | N/A |
| pCMV10 3xFlag C101A CISD2 peptide | This study | N/A |
| pCMV10 3xFlag 16 a.a. CISD2 peptide | This study | N/A |
| pCMV10 3xFlag 14 a.a. CISD2 peptide | This study | N/A |
| **Antibodies** | | |
| Rabbit anti-Flag | Cell Signaling | 2368S |
| Mouse anti-Flag | MBL | M185-3L |
| Mouse anti-Myc | MBL | M192-3 |
| Mouse anti-GFP | Santa Cruz | sc-9996 |
| Rabbit anti-WFS1 | Cell Signaling | 8749S |
| Rabbit anti-CISD2 | Proteintech | 13318-1-AP |
| Rabbit anti-IP$_3$R1 | Proteintech | 19962-1-AP |
| Rabbit anti-BCL2 | Cell Signaling | 4223T |
| Rabbit anti-HA | Cell Signaling | 3724S |
| Mouse anti-tubulin | DSHB | N/A |
| HRP-mouse | Jackson ImmunoResearch | 115-035-146 |
| HRP-rabbit | Jackson ImmunoResearch | 111-035-144 |
| Anti-DILP2 antibody | Dr. Kweon Yu (University of Science and Technology, Korea) | N/A |
| Anti-rabbit TRITC | Jackson ImmunoResearch | 111-296-144 |
| **Oligonucleotides and other sequence-based reagents** | | |
| BCL2 siRNA | Bioneer | 596-3 (Sequence information is available in the Methods section) |
| CPP-rhodamine B peptide | SP$^2$ Therapeutics | N/A |
| WT CISD2-CPP-rhodamine B peptide | SP$^2$ Therapeutics | N/A |
| sgRNA | This study | Methods section |
| qPCR primer | This study | Appendix Table S1 |
| **Chemicals, Enzymes and other reagents** | | |
| DMEM | Welgene | LM 001-05 |
| FBS | Gibco | 16000044 |
| Polyethyleneimine | Sigma | 408727 |
| Lipofectamine LTX | Invitrogen | 15338100 |
| Lipofectamine 2000 | Invitrogen | 11668019 |
| Lipofectamine 3000 | Invitrogen | L3000150 |
| IP$_3$ | Sigma | 850115P |
| Glucose assay reagent | Sigma | G3293 |

| Reagent/Resource | Reference or Source | Identifier or Catalog Number |
|---|---|---|
| Pierce BCA protein assay kit | Thermo Scientific | 23225 |
| Lipoprotein lipase | Calbiochem | 437707 |
| Free glycerol reagent | Sigma | F6428 |
| TRIzol | Invitrogen | 15596018 |
| Random primer | Promega | C1181 |
| M-MLV reverse transcriptase | Promega | M1701 |
| TOPreal SYBR Green qPCR PreMIX | Enzynomics | RT500M |
| Brilliant blue FCF | Merck | 80717 |
| Millex filter | Millipore | SLMP025SS |
| ATP | Sigma | A3377 |
| Fibronectin bovine plasma | Sigma | F1141 |
| β-escin | Sigma | E1378 |
| **Software** | | |
| ImageJ | National Institutes of Health | N/A |
| GraphPad Prism v.10 | GraphPad Software | N/A |
| Zen software | Carl Zeiss | N/A |
| NIS-Elements Advanced Research software | Nikon | N/A |
| Multi Gauge V3.0 | Fujifilm Life Science | N/A |

## Plasmid constructs and siRNAs

pcDNA3 WFS1-Flag (NM_001145853.1) was purchased from Addgene. WFS1, WFS1 ΔN mutant (311–890 amino acid) and WFS1 ΔC mutant (1–654 amino acid) were cloned into a pcDNA3 HA vector. CISD2 (NM_001008388.5) was a gift from Dr. Jae Ung Jung (Harvard Medical School, USA). CISD2 and CISD2 D1 mutant (1–99 amino acid) were cloned into a pcDNA3.1 zeo (+) C-terminal Myc/His-tagged vector. CISD2 C99A, C101A, C110A, and H114A mutants were generated using a site-directed point mutagenesis method. GFP-IP$_3$R1 (NM_174841.2), Flag-IP$_3$R1 truncated mutants (SD, LBS, MD1, MD2, MD3, and GK), and IP$_3$R1 cloned into a pcDNA3 vector were gifts from Dr. Sang Ki Park (Postech, Korea) (Park et al, 2017). Two single-stranded oligonucleotides with complementary sequences of WT CISD2 peptide (99–116 amino acid) were annealed and cloned into pCMV10 N-terminal Flag-tagged vector. The C101A CISD2 peptide, 16 amino acid CISD2 peptide (99–114 amino acid), and 14 amino acid CISD2 peptide (101–114 amino acid) were generated using a site-directed point mutagenesis method. All of the cloning and mutagenesis experiments were performed using DH10β Escherichia coli. The BCL2 siRNA was purchased from Bioneer, Korea. The BCL2 siRNA sequences used in immunoblot analysis are as shown below:

BCL2 siRNA(AS) GAGAUAGUGAUGAAGUACA=tt
BCL2 siRNA(AA) UGUACUUCAUCACUAUCUC=tt

## Cell-penetrating peptides and their treatment

The synthesized cell-penetrating peptides (CPP-rhodamine B and WT CISD2-CPP-rhodamine B) were obtained from SP² Therapeutics, Korea. HEK293 WT, HEK293 WFS1 KO, and HEK293 CISD2 KO cells were cultured on 10 mm fibronectin bovine plasma-coated coverslips embedded in a 24-well plate. Cells were then incubated with 10 μM of the designated cell-penetrating peptides in KRB Buffer [140 mM NaCl, 3.6 mM KCl, 0.5 mM $NaH_2PO_4$, 0.5 mM $MgSO_4$, 1.5 mM $CaCl_2$, 10 mM HEPES, 2 mM $NaHCO_3$, 5.5 mM glucose, and pH 7.4-titrated with NaOH] at 37 °C for 1 h followed by washing twice with KRB buffer. Live cells were observed and monitored for peptide uptake visualization and calcium measurement using LSM710 laser scanning confocal microscope (Carl Zeiss, Germany).

## Cell culture and transfection

HEK293 WT, HEK293 WFS1 KO, HEK293 CISD2 KO, and HEK293T cells were used. Human Embryonic Kidney 293 (HEK293) and HEK293T cell lines were gifted from Dr. John Blenis at Cornell University. All cell lines were authenticated and tested for mycoplasma contamination. All cell lines were cultured in DMEM (Welgene, Korea) supplemented with 10% fetal bovine serum (Gibco) at 37 °C in a humidified atmosphere composed of 5% $CO_2$. HEK293T cells were transfected using a polyethyleneimine reagent (Sigma). HEK293 cells were transfected using Lipofectamine LTX, 2000, or 3000 as instructed by the manufacturer (Invitrogen).

## Generation of WFS1 KO and CISD2 KO cells

The CRISPR genome editing technique was used for the generation of WFS1 or CISD2 lacking cells. To generate WFS1 KO HEK293 cells, the guide RNA sequence (CCCTGGTGTTAGAGACGCAG) was cloned into the PX459 vector (Addgene, #62988) (Ran et al, 2013). The plasmid was transfected into HEK293 cells. 48 h after transfection, transfected cells were selected by 10 μg/ml puromycin for 3 days, and then single colonies were transferred onto 96-well plates with one colony in each well. The WFS1 KO clones were screened by immunoblot analysis with rabbit anti-WFS1 antibody (1:1000, Cell Signaling). The guide RNA sequence (GAGCGTGGCCCGTATCGTGA) was used to generate CISD2 KO HEK293 cells. CISD2 KO clones were generated as described above and screened by immunoblot analysis with rabbit anti-CISD2 antibody (1:1000, Proteintech).

## Antibodies

The following antibodies were used: rabbit anti-Flag (IB, 1:1000, Cell Signaling, 2368S, polyclonal), mouse anti-Flag (IP, 1:500; MBL, M185-3L, monoclonal (FLA-1)), mouse anti-Myc (IP, 1:500; IB, 1:1000; MBL, M192-3, monoclonal (My3)), mouse anti-GFP (IB, 1:1000; Santa Cruz, sc-9996, monoclonal (B-2)), rabbit anti-WFS1 (IP, 1:2000; IB, 1:1000; Cell Signaling, 8749S, polyclonal), rabbit anti-CISD2 (IB, 1:1000; Proteintech, 13318-1-AP, polyclonal), rabbit anti-IP₃R1 (IB, 1:1000; Proteintech, 19962-1-AP,

polyclonal), rabbit anti-BCL2 (IB, 1:1000; Cell Signaling, 4223T, monoclonal (D55G8)), rabbit anti-HA (IP, 1:500; IB, 1:1000; Cell Signaling, 3724S, monoclonal (C29F4)), and mouse anti-tubulin (IB, 1:10,000; Developmental Studies Hybridoma Bank, monoclonal (E7)). Peroxidase-conjugated secondary antibodies were used: HRP-mouse (WB, 1:5000; Jackson ImmunoResearch, 115-035-146, polyclonal) and HRP-rabbit (WB, 1:5000; Jackson ImmunoResearch, 111-035-144, polyclonal).

## Immunoprecipitation and immunoblotting

To perform immunoprecipitation, cells were lysed using a lysis buffer [20 mM tris (pH 7.5), 100 mM NaCl, 1 mM EDTA, 2 mM EGTA, 50 mM β-glycerophosphate, 50 mM NaF, 1 mM sodium vanadate, 1 mM phenylmethylsulfonyl fluoride (PMSF), leupeptin (10 μg/ml), pepstatin A (1 μg/ml), and 1% Triton X-100]. The cell lysates were centrifuged at 13,000 rpm at 4 °C for 20 min. Then, the supernatant was incubated at 4 °C overnight after the addition of primary antibodies. The lysates were incubated with protein A/G agarose beads for 2 h at 4 °C, washed four times in detergent-free lysis buffer, and eluted with 2X Laemmli buffer at 95 °C. To perform immunoblotting, flies or cells were lysed using radio-immunoprecipitation assay (RIPA) buffer [50 mM tris (pH 8.0), 150 mM NaCl, 0.5% sodium deoxycholate, 1% NP-40, 0.1% SDS, 50 mM NaF, 1 mM sodium vanadate, 1 mM PMSF, leupeptin (10 μg/ml), and pepstatin A (1 μg/ml)]. Total protein was quantified using Pierce BCA protein assay kit (#23225, Thermo Scientific). The samples were subjected to SDS-PAGE analysis. LAS-4000 (Fujifilm) was used to develop and observe the immunoblots. Immunoblot band intensity was quantified using ImageJ.

## Dot blot analysis

To perform dot blotting, flies or cells were lysed using a lysis buffer [20 mM tris (pH 7.5), 100 mM NaCl, 1 mM EDTA, 2 mM EGTA, 50 mM β-glycerophosphate, 50 mM NaF, 1 mM sodium vanadate, 1 mM phenylmethylsulfonyl fluoride (PMSF), leupeptin (10 μg/ml), pepstatin A (1 μg/ml), and 1% Triton X-100]. The lysates were centrifuged at 13,000 rpm at 4 °C for 20 min. Then, the supernatant was incubated at 4 °C overnight after the addition of primary antibodies. The lysates were incubated with protein A/G agarose beads for 2 h at 25 °C, washed four times in detergent-free lysis buffer, and eluted with a dilution buffer [8 M urea, 100 mM $NaH_2PO_4$, 10 mM Tris-Cl, and pH 8.0-titrated with HCl]. Eluted samples were loaded onto 0.2 μM nitrocellulose membrane. Dried membrane was blocked with blocking solution [4% BSA, 50 mM Tris, 0.5 M NaCl, 0.05% Tween-20, and pH 7.4] for 1 h at room temperature. Afterwards, the membrane was incubated with a primary antibody at 4 °C overnight, washed four times in TTBS [50 mM Tris, 0.5 M NaCl, 0.05% Tween-20, and pH 7.4], incubated with a secondary antibody for 1 h at room temperature, and washed six times in TTBS. LAS-4000 (Fujifilm) was used to develop and observe the dot blots.

## Measurement of calcium levels in mammalian cells

HEK293 WT, HEK293 WFS1 KO, and HEK293 CISD2 KO cells were cultured on 10 mm fibronectin bovine plasma-coated coverslips embedded in a 24-well plate, and transfected with ER calcium

indicator G-CEPIA1er (472 ± 15 nm excitation/520 ± 17.5 nm emission; Addgene, #105012) or cytosol calcium indicator RCaMP1h (543 ± 20 nm excitation/580 ± 20 nm emission; Addgene, #105014) using Lipofectamine 3000. After 48 h of transfection, we monitored the live cells expressing G-CEPIA1er and RCaMP1h using LSM710 laser scanning confocal microscope (Carl Zeiss, Germany) and eclipse ti2 microscope (Nikon, Japan), respectively. The cells were placed in a 37 °C heated chamber and perfused with KRB buffer [140 mM NaCl, 3.6 mM KCl, 0.5 mM $NaH_2PO_4$, 0.5 mM $MgSO_4$, 1.5 mM $CaCl_2$, 10 mM HEPES, 2 mM $NaHCO_3$, 5.5 mM glucose, and pH 7.4-titrated with NaOH]. After 2–3 min of baseline recording, a single pulse of 100 μM ATP was delivered to liberate calcium stores for 3 min and then washed out. Peak amplitudes of $Ca^{2+}$ responses to 100 μM ATP were normalized to the basal fluorescence ($F_0$) before stimulation. The area-over-the-curve (AOC) of ER calcium release and area-under-the-curve (AUC) of cytosolic calcium levels were calculated by multiplying the changes in fluorescence over the basal ($\Delta F/F_0$) by the time (sec). Calcium transients were continuously recorded and analyzed on Zen software (Carl Zeiss, Germany) or NIS-Elements Advanced Research software (Nikon, Japan).

## Measurement of calcium levels in flies

ERGCaMP6-210 was expressed in the muscle using *mef2-Gal4* and UAS-ERGCaMP6-210 for ER calcium measurement in *Drosophila* larval muscle. GCaMP5G was expressed using *mef2-Gal4* and UAS-GCaMP5G for cytosolic calcium measurement. Larvae were dissected in a perfusion buffer [2 mM $CaCl_2$, 4 mM $MgCl_2$, 2 mM KCl, 2 mM NaCl, 5 mM HEPES, 35.5 mM sucrose, 7 mM L-glutamic acid, and pH 7.3-titrated with NaOH] on a stereomicroscope. After dissection, larval muscles were monitored using Eclipse Ti2 microscope (Nikon, Japan). The larvae were placed in a chamber and perfused with a perfusion buffer. After 2–3 min of baseline recording, a single pulse of 10 mM ATP was delivered to release calcium stores for 3 min, followed by a washout. Peak amplitudes of calcium responses to 10 mM ATP were normalized to the basal fluorescence ($F_0$) before stimulation. The area-over-the-curve (AOC) and area-under-the-curve (AUC) in the bar graphs were calculated by multiplying the changes in fluorescence over the basal ($\Delta F/F_0$) by the time (s). Calcium transients were continuously recorded and analyzed on NIS-Elements Advanced Research software (Nikon, Japan).

## Measurement of the influx and efflux of ER calcium in mammalian cells

For this assay, 40 μM β-escin in intracellular medium [ICM; 10 mM HEPES, 125 mM KCl, 19 mM NaCl, 1 mM EGTA, and pH 7.3-titrated with KOH] was used for 100 s to permeabilize cells. After washing with ICM for 5 min, permeabilized cells were superfused for 3–4 min with loading buffer [10 mM HEPES, 125 mM KCl, 19 mM NaCl, 1 mM EGTA, 0.65 mM $CaCl_2$, 1.4 mM $MgCl_2$, 3 mM $Na_2ATP$, and pH 7.3-titrated with KOH] to stimulate SERCA and load $Ca^{2+}$ stores. After that, release buffer [10 mM HEPES, 125 mM KCl, 19 mM NaCl, 1 mM EGTA, 0.65 mM $CaCl_2$, and pH 7.3-titrated with KOH] with 0.5 μM $IP_3$ (850115P, Sigma) was superfused for 3–4 min to stimulate $IP_3R$. Peak amplitudes of $Ca^{2+}$ responses to solution changes were normalized to the basal fluorescence ($F_0$) before stimulations. The ER $Ca^{2+}$ release rate of

the bar graph was calculated by dividing the changes in fluorescence over the maximum ($\Delta F/F_{max}$) by the time (sec).

## Fly stocks and maintenance

The *Drosophila* lines used in our experiments are below: Bloomington Drosophila Stock Center: *w1118* (3605), *tub-Gal4* (5138), UAS-*Itpr* (30742), UAS-*WFS1* (8357), UAS-*LacZ* (1776), *mef2-Gal4* (27390), UAS-ERGCaMP6-210 (83294), and UAS-GCaMP5G (42037). *dCISD* KO flies were generated in a previous study (Ham et al, 2023). Flies were grown and aged on food containing 35 g cornmeal, 70 g dextrose, 5 g agar, 50 g dry active yeast, 4.6 ml propionic acid, and 7.3 ml Tegosept (100 g/l in ethanol) per liter at 25 °C on a 12 L/12D cycle. An equal number of female and male flies was used for all fly samples.

## Generation of *WFS1* KO, UAS-*dCISD*, and UAS-dCISD peptide flies

To generate *WFS1* KO flies, we injected *w1118* fly embryos with a pU6-Bbs1-chiRNA vector (#45946, Addgene) containing the 20-bp sgRNA sequence (GCTCACCAGGCGTCATAGCC), along with the Cas9 expression vector (pHsp70-Cas9; #45945, Addgene). After the injection, *WFS1* KO flies were sorted through PCR and subsequent DNA sequencing. To generate UAS-*dCISD* flies, we injected *w1118* fly embryos with a pUAST N-terminal Flag-tagged vector with d*CISD* insertion and sorted flies. Expression was confirmed by crossing candidate lines with the ubiquitous *tub-Gal4* driver and confirming through immunoblot assay. To generate UAS-dCISD peptide flies, we injected *w1118* fly embryos with a pUAST vector with dCISD peptide-Flag insertion and sorted. Expression was confirmed by crossing candidate lines with the ubiquitous *tub-Gal4* driver, after which the RNA of progeny was collected and assayed for dCISD peptide mRNA levels through qPCR analysis using the primers listed in Appendix Table S1.

## Hemolymph glucose measurements

Thirty flies were punctured with a microneedle in the thorax and transferred to a 500 μl tube with small holes punctured in the bottom, which were centrifuged at 3000 rpm for 5 min at 4 °C to collect 1 μl of hemolymph. The collected hemolymph was diluted 1:10, and 10 μl of diluted hemolymph was mixed with 200 μl of glucose assay reagent (G3293, Sigma) and incubated at 37 °C for 10 min. Then, absorbance at 340 nm was measured using Tecan Plate Reader Infinite 200. Thirty flies per genotype were collected for a total of 10 samples per genotype, conducted across at least 3 independent experiments.

## TAG measurements

10 flies were homogenized in 500 μl of 0.1% PBST (1X PBS with 0.1% Tween 20) using plastic pestles. 10 μl of the lysate was used to measure the total protein levels by Pierce BCA protein assay kit (23225, Thermo Scientific). The remaining 490 μl of the lysate was incubated at 70 °C for 5 min and then chilled on ice for 10 min. 1 μl of lipoprotein lipase from *Chromobacterium viscosum* (437707, Calbiochem) was added to the lysate and incubated at 37 °C overnight. Then, the lysate was centrifuged at 14,000 rpm for 10 min, and 20 μl of the resulting supernatant was mixed with 180 μl of free glycerol reagent (F6428,

Sigma) and incubated at 37 °C for 10 min. The absorbance at 540 nm of the samples was measured using Tecan Plate Reader Infinite 200, and resulting measurements were divided by their respective protein levels to normalize the data. 10 flies per genotype were collected for a total of 10 samples per genotype, conducted across at least 3 independent experiments.

### *Drosophila* glucose tolerance test (GTT)

Flies were fasted for 1 day, fed a 10% sucrose solution, then fasted again during which hemolymph glucose levels were monitored. Hemolymph glucose was measured before and directly after starvation, after the 10% sucrose feeding, and 1.5 h and 3 h during the subsequent fasting. 30 flies per genotype were collected at each time point for measurement, and repeated independently 3 times.

### RNA extraction and real-time PCR

Ten flies were homogenized in 500 µl of TRIzol solution (15596018, Invitrogen) and centrifuged at 14,000 rpm at 4 °C for 15 min. The supernatant was mixed with 100 µl of chloroform and centrifuged at 14,000 rpm at 4 °C for 15 min. Then, 200 µl of the transparent solution was mixed with 200 µl of isopropanol and centrifuged at 14,000 rpm at 4 °C for 15 min. The supernatant was then removed and the remaining pellet was washed by using 500 µl of 75% ethanol and centrifuging at 14,000 rpm at 4 °C for 15 min. The supernatant was again removed and the pellet was dried, after which the pellet was dissolved in 10 µl of RNase free water. 1 µl of random primer (C1181, Promega) was added to 10 µl of RNA solution and incubated at 70 °C for 10 min, after which the solution was cooled down. M-MLV reverse transcriptase (M1701, Promega) was used to synthesize cDNA. Real-time PCR was performed using TOPreal SYBR Green qPCR PreMIX (RT500M, Enzynomics). Gene expression was normalized with *rp49* measurements. 10 flies per genotype were collected for a total of 10 samples per genotype, conducted across at least 3 independent experiments. The primer sequences used in the real-time PCR experiments are listed in Appendix Table S1.

### DILP2 immunostaining

Flies were fixed in 4% paraformaldehyde for 3 h, after which brains were dissected. Brains were washed and permeabilized with 0.1% PBST [1X PBS with 0.1% Tween-20] three times for 10 min each. 0.1% PBST solution with 3% BSA was used to block brains for one hour. Brains were again washed with 0.1% PBST, then treated with anti-DILP2 antibody (Kwak et al, 2013) at 1:500 at 4 °C overnight. Samples were washed with 0.1% PBST and incubated at RT for 1 h with anti-rabbit TRITC (111-296-144, Jackson) as the secondary antibody. The brains were mounted in 80% PBG [1X PBS with 80% glycerol] and observed using the LSM710 confocal microscope (Carl Zeiss) and visualized using Z-stack analysis. Fluorescence intensity was quantified using ImageJ, for a total of 10 brains per genotype assayed across at least 3 independent experiments.

### Food intake assay

Flies were starved for 24 h and provided only with water, then fed a solution of 10% sucrose and 1% brilliant blue FCF (80717, Merck) for 1 h. 10 flies per sample were then homogenized in 200 µl of

distilled water and an additional 800 µl of distilled water was added, after which the mixture was filtered through a 0.22 µm Millex filter (SLMP025SS, Millipore). The absorbance of the filtered homogenates was measured at 629 nm using Tecan Plate Reader. 10 flies per genotype were collected for a total of 10 samples per genotype, conducted across at least 3 independent experiments.

### Statistical analysis

A blind manner was used in all experiments and analyses. Image areas were randomly selected during observing samples. For computing *p*-values, one-way ANOVA (Tukey's multiple-comparison test) was used. All tests were examined via GraphPad Prism v.10 (GraphPad Software) for the statistics.

## Data availability

This study includes no data deposited in external repositories. All data are included in the manuscript and expanded view information.

The source data of this paper are collected in the following database record: biostudies:S-SCDT-10_1038-S44319-025-00436-2.

## Peer review information

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

## Acknowledgements

This study was supported by the National Research Foundation of Korea and grants were funded by the Korean government (MSIT) (RS-2020-NR049538 and RS-2024-00339984). SJH was supported by the National Research Foundation of Korea (RS-2023-00211029). JC, EY, DHL, and SK were supported by the BK21 Plus Program from the Ministry of Education, Korea. We would like to thank Flybase and Bloomington Drosophila Stock Center for fly stocks and related information.

## Author contributions

**Su Jin Ham**: Conceptualization; Resources; Data curation; Formal analysis; Validation; Investigation; Visualization; Methodology; Writing—original draft; Writing—review and editing. **Eunju Yoon**: Conceptualization; Resources; Data curation; Formal analysis; Validation; Investigation; Visualization; Methodology; Writing—original draft; Writing—review and editing. **Da Hyun Lee**: Conceptualization; Resources; Data curation; Formal analysis; Validation; Investigation; Visualization; Methodology; Writing—original draft; Writing—review and editing. **Sehyeon Kim**: Resources; Formal analysis; Validation; Investigation; Visualization; Methodology. **Heesuk Yoo**: Resources. **Jongkyeong Chung**: Conceptualization; Resources; Data curation; Software; Formal analysis; Supervision; Funding acquisition; Validation; Investigation; Visualization; Methodology; Writing—original draft; Project administration; Writing—review and editing.

Source data underlying figure panels in this paper may have individual authorship assigned. Where available, figure panel/source data authorship is listed in the following database record: biostudies:S-SCDT-10_1038-S44319-025-00436-2.

## Disclosure and competing interests statement

The authors applied for patents from the data obtained from this research.

# Expanded View Figures

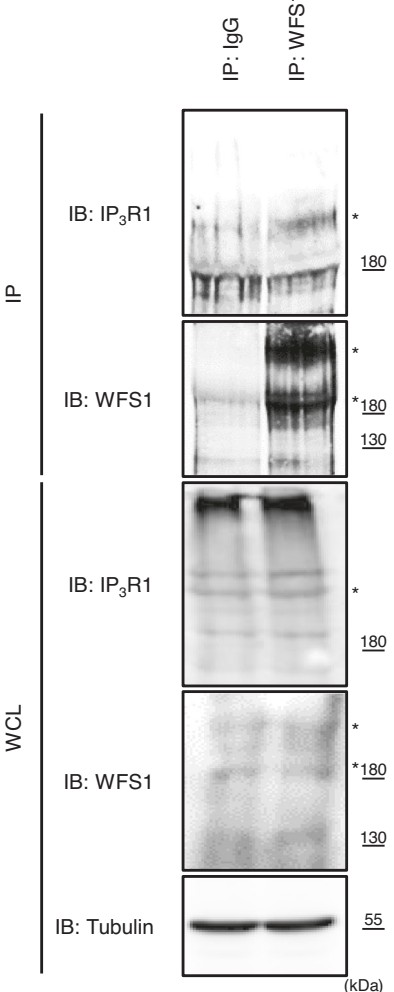

**Figure EV1. WFS1 interacts with IP₃R1 at the endogenous level.**

HEK293 cell lysates were subjected to anti-IgG or anti-WFS1 immunoprecipitation followed by immunoblot analysis. The asterisks denote the band of interest.

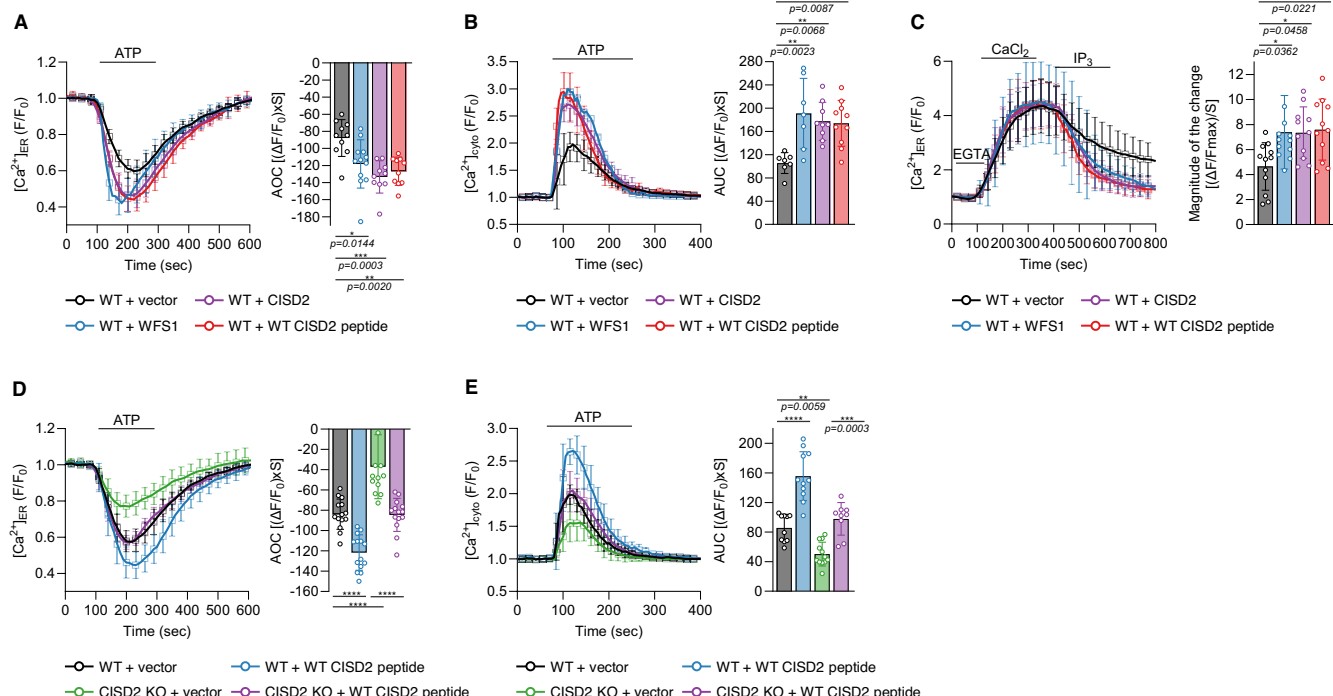

**Figure EV2. IP₃R activity is elevated by overexpression of WFS1, CISD2, or CISD2 peptide.**

(A) Measurement of ER calcium modulation in WT HEK293 cells transfected with empty vector (black, $n = 73$ cells, 10 coverslips), *WFS1* (blue, $n = 54$ cells, 12 coverslips), *CISD2* (purple, $n = 59$ cells, 10 coverslips) or *WT CISD2 peptide* (red, $n = 50$ cells, 10 coverslips). (B) Measurement of cytosolic calcium modulation in WT HEK293 cells transfected with empty vector (black, $n = 113$ cells, 7 coverslips), *WFS1* (blue, $n = 105$ cells, 7 coverslips), *CISD2* (purple, $n = 109$ cells, 9 coverslips) or *WT CISD2 peptide* (red, $n = 126$ cells, 10 coverslips). The right-side bar graphs indicate the quantification of the normalized calcium traces using AOC or AUC of calcium fluxes during ATP treatment. (C) Measurement of the IP₃R activity of WT HEK293 cells transfected with empty vector (black, $n = 91$ cells, 12 coverslips), *WFS1* (blue, $n = 53$ cells, 12 coverslips), *CISD2* (purple, $n = 72$ cells, 11 coverslips), or *WT CISD2 peptide* (red, $n = 50$ cells, 11 coverslips). The right-side bar graph represents the magnitude of the change during IP₃ treatment. (D) Measurement of ER calcium modulation in WT HEK293 cells transfected with empty vector (black, $n = 129$ cells, 14 coverslips) or *WT CISD2 peptide* (blue, $n = 117$ cells, 14 coverslips) and CISD2 KO HEK293 cells transfected with empty vector (green, $n = 146$ cells, 13 coverslips) or *WT CISD2 peptide* (purple, $n = 133$ cells, 14 coverslips). (E) Measurement of cytosolic calcium modulation in WT HEK293 cells transfected with empty vector (black, $n = 119$ cells, 10 coverslips) or *WT CISD2 peptide* (blue, $n = 98$ cells, 11 coverslips) and CISD2 KO HEK293 cells transfected with empty vector (green, $n = 113$ cells, 12 coverslips) or *WT CISD2 peptide* (purple, $n = 119$ cells, 9 coverslips). The right-side bar graphs indicate the quantification of the normalized calcium traces using AOC or AUC of calcium fluxes during ATP treatment. Data information: All figures are representatives of three or more independent experiments. All quantifications were analyzed by one-way ANOVA with Tukey multiple-comparison test. *$p < 0.05$, **$p < 0.01$, ***$p < 0.001$, ****$p < 0.0001$. All data are presented as mean ± SD.

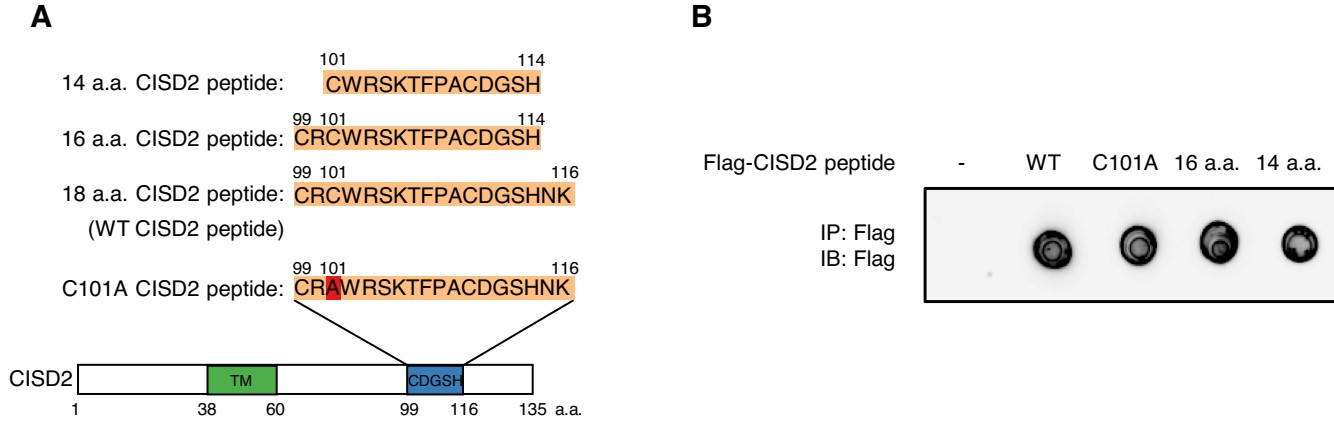

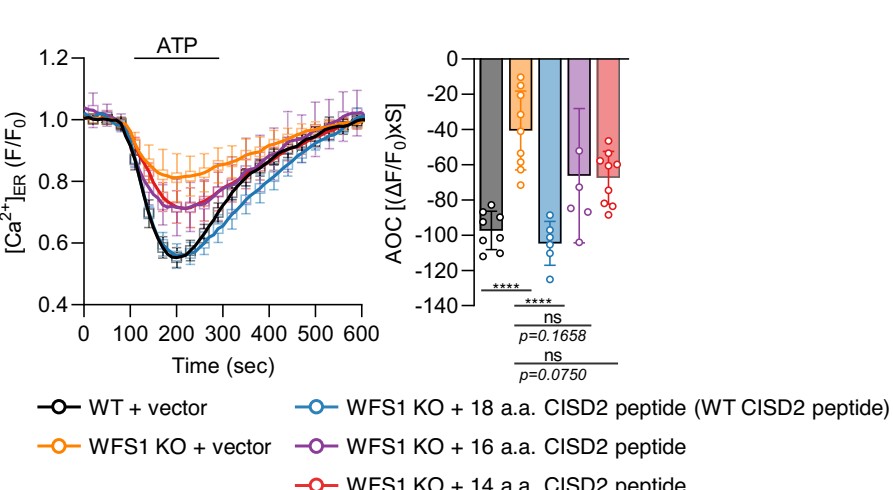

**Figure EV3.   Overexpression of 18 amino acid CISD2 peptide, but not 16 and 14 amino acid CISD2 peptide, rescues altered calcium modulation in WFS1 knockout cells.**

(A) Schematic diagram showing various forms of CISD2 peptides. (B) HEK293T cells were transfected as indicated and cell lysates were subjected to anti-Flag immunoprecipitation followed by dot blot analysis. (C) Measurement of ER calcium release in WT HEK293 cells transfected with empty vector (black, $n = 60$ cells, 8 coverslips) and WFS1 KO HEK293 cells transfected with empty vector (orange, $n = 81$ cells, 9 coverslips), *18 amino acid CISD2 peptide* (referred to as WT CISD2 peptide) (blue, $n = 50$ cells, 6 coverslips), *16 amino acid CISD2 peptide* (purple, $n = 60$ cells, 6 coverslips) or *14 amino acid CISD2 peptide* (red, $n = 72$ cells, 9 coverslips). The right-side bar graphs indicate the quantification of the normalized calcium traces using AOC of calcium fluxes during ATP treatment. Data information: All figures are representatives of three or more independent experiments. All quantifications were analyzed by one-way ANOVA with Tukey multiple-comparison test. ****$p < 0.0001$. ns, not significant. All data are presented as mean ± SD.

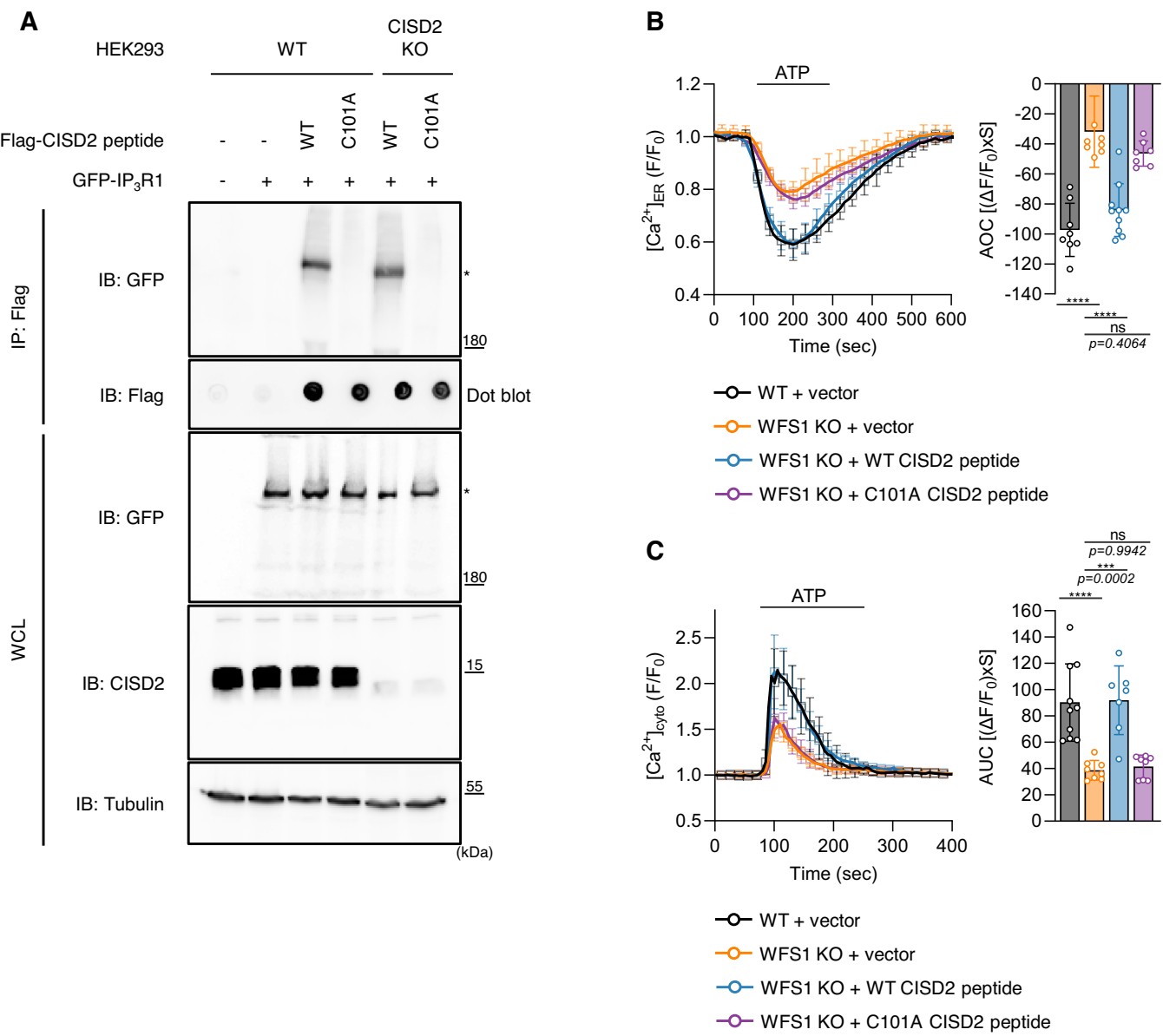

**Figure EV4. WT CISD2 peptide, but not C101A CISD2 peptide, directly binds to IP$_3$R1 and rescues altered calcium modulation in WFS1 knockout cells.**

(A) WT and CISD2 KO HEK293 cells were transfected as indicated and cell lysates were subjected to anti-Flag immunoprecipitation followed by immunoblot analysis or dot blot analysis. (B) Measurement of ER calcium modulation in WT HEK293 cells transfected with empty vector (black, $n = 86$ cells, 8 coverslips) and WFS1 KO HEK293 cells transfected with empty vector (orange, $n = 94$ cells, 8 coverslips), *WT CISD2 peptide* (blue, $n = 116$ cells, 10 coverslips), or *C101A CISD2 peptide* (purple, $n = 98$ cells, 7 coverslips). (C) Measurement of cytosolic calcium modulation in WT HEK293 cells transfected with empty vector (black, $n = 90$ cells, 10 coverslips) and WFS1 KO HEK293 cells transfected with empty vector (orange, $n = 65$ cells, 8 coverslips), *WT CISD2 peptide* (blue, $n = 96$ cells, 7 coverslips), or *C101A CISD2 peptide* (purple, $n = 91$ cells, 8 coverslips). The right-side bar graphs indicate the quantification of the normalized calcium traces using AOC or AUC calcium fluxes during ATP treatment. Data information: All figures are representatives of three or more independent experiments. All quantifications were analyzed by one-way ANOVA with Tukey multiple-comparison test. ***$p < 0.001$, ****$p < 0.0001$. ns, not significant. All data are presented as mean ± SD.

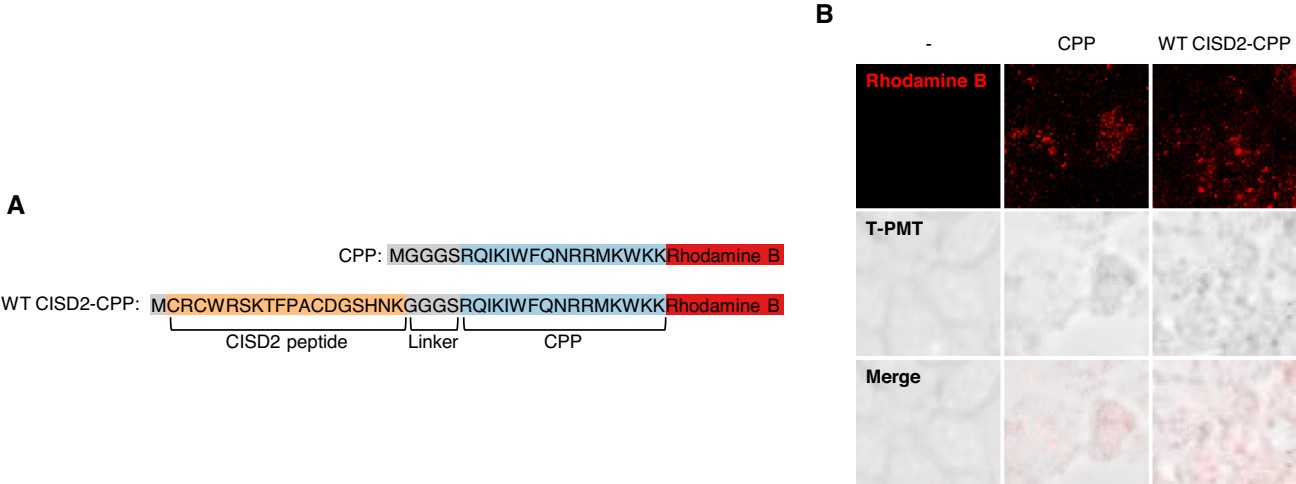

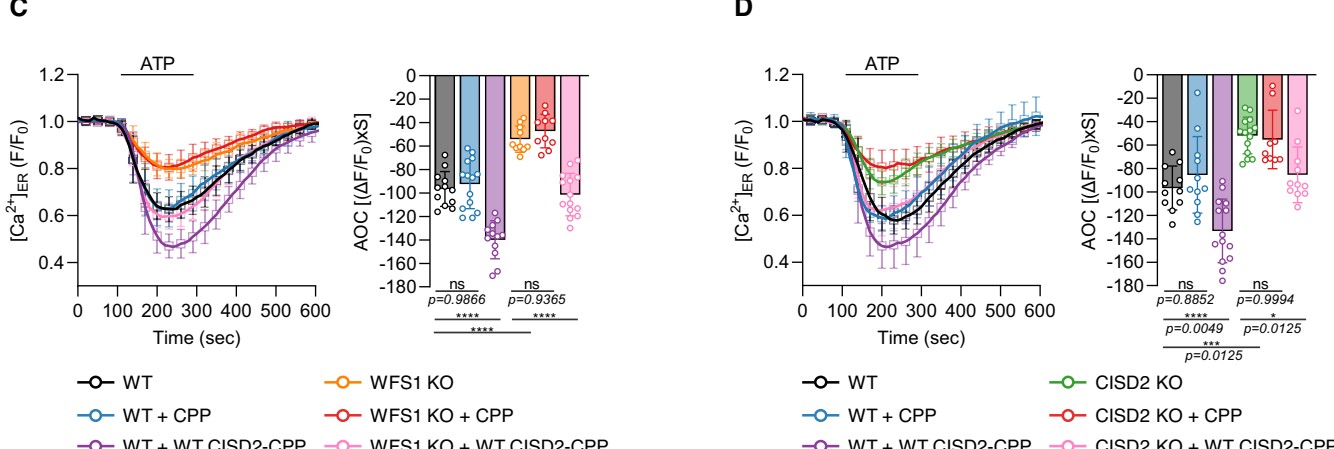

**Figure EV5. Treatment of CISD2 cell-penetrating peptide rescues altered calcium modulation in WFS1 or CISD2 knockout cells.**

(A) Schematic diagram representing cell-penetrating peptide (CPP) and WT CISD2-CPP. (B) Representative images of cells incubated with or without cell-penetrating peptides. Peptide uptake was visualized via Rhodamine B fluorescence (red), transmitted-photomultiplier tube (T-PMT) as brightfield, and merged images. Scale bar, 10 μm. (C) ER calcium modulation of WT HEK293 cells was measured after incubation with pure buffer (black, $n = 96$ cells, 12 coverslips), 10 μM CPP (blue, $n = 97$ cells, 14 coverslips), or 10 μM WT CISD2-CPP (purple, $n = 92$ cells, 12 coverslips) at 37 °C for 1 h. ER calcium modulation of WFS1 KO cells was measured after incubation with pure buffer (orange, $n = 107$ cells, 10 coverslips), 10 μM CPP (red, $n = 95$ cells, 10 coverslips), or 10 μM WT CISD2-CPP (pink, $n = 112$ cells, 14 coverslips) at 37 °C for 1 h. (D) ER calcium modulation of WT HEK293 cells was measured after incubation with pure buffer (black, $n = 107$ cells, 11 coverslips), 10 μM CPP (blue, $n = 85$ cells, 11 coverslips), or 10 μM WT CISD2-CPP (purple, $n = 83$ cells, 14 coverslips) at 37 °C for 1 h. ER calcium modulation of CISD2 KO cells was measured after incubation with pure buffer (green, $n = 84$ cells, 14 coverslips), 10 μM CPP (red, $n = 93$ cells, 9 coverslips), or 10 μM WT CISD2-CPP (pink, $n = 117$ cells, 11 coverslips) at 37 °C for 1 h. The right-side bar graphs indicate the quantification of the normalized calcium traces using AOC of calcium fluxes during ATP treatment. Data information: All figures are representatives of three or more independent experiments. All quantifications were analyzed by one-way ANOVA with Tukey multiple-comparison test. $*p < 0.05$, $***p < 0.001$, $****p < 0.0001$. ns, not significant. All data are presented as mean ± SD.

