## [Peer Review File · EMBO Reports]

Reciprocal rescue of Wolfram syndrome by two causative genes

Jongkyeong Chung, Su Jin Ham, Eunju Yoon, Da Hyun Lee, Sehyeon Kim, and Heesuk Yoo

Corresponding author(s): Jongkyeong Chung (jkc@snu.ac.kr)

Review Timeline:

Submission Date:	28th Feb 24
Editorial Decision:	12th Apr 24
Revision Received:	3rd Jan 25
Editorial Decision:	6th Feb 25
Revision Received:	24th Feb 25
Accepted:	7th Mar 25

Editor: Deniz Senyilmaz Tiebe

Transaction Report:

Dear Prof. Chung,

Thank you for the submission of your research manuscript to our journal, which was now seen by two referees, whose reports are copied below.

My apologies for this unusual delay in getting back to you. It took longer than anticipated to receive the full set of referee reports.

Referees express interest in the proposed interplay between WFS1 and CISD2 in ER calcium regulation and WS related diabetic phenotypes. However, they also raise significant concerns that need to be addressed to consider publication here.

I find the reports informed and constructive, and believe that addressing the concerns raised will significantly strengthen the manuscript. As the reports are below, and I think all points need to be addressed, I will not detail them here.

Given these recommendations, we would like to invite you to submit a revised manuscript. Please revise your manuscript with the understanding that the referee concerns (as in their reports) must be fully addressed and their suggestions taken on board. Please address all referee concerns in a complete point-by-point response. Acceptance of the manuscript will depend on a positive outcome of a second round of review. It is EMBO reports policy to allow a single round of major experimental revision only and acceptance or rejection of the manuscript will therefore depend on the completeness of your responses included in the next, final version of the manuscript.

We realize that it is difficult to revise to a specific deadline. In the interest of protecting the conceptual advance provided by the work, we recommend a revision within 3 months. Please discuss the revision progress ahead of this time with me if you require more time to complete the revisions, or if you have questions or comments regarding the revision (also by video chat).

1. A data availability section providing access to data deposited in public databases is missing (where applicable).
2. Your manuscript contains statistics and error bars based on $n=2$. Please use scatter plots in these cases.

You can submit the revision either as a Scientific Report or as a Research Article. For Scientific Reports, the revised manuscript can contain up to 5 main figures and 5 Expanded View figures, and it should not exceed 27000 characters. If the revision leads to a manuscript with more than 5 main figures it will be published as a Research Article. In this case the Results and Discussion section should be separate. If a Scientific Report is submitted, these sections have to be combined. This will help to shorten the manuscript text by eliminating some redundancy that is inevitable when discussing the same experiments twice. In either case, all materials and methods should be included in the main manuscript file.

3) We replaced Supplementary Information with Expanded View (EV) Figures and Tables that are collapsible/expandable online. A maximum of 5 EV Figures can be typeset. EV Figures should be cited as 'Figure EV1, Figure EV2' etc... in the text and their respective legends should be included in the main text after the legends of regular figures.

4) a .docx formatted letter INCLUDING the reviewers' reports and your detailed point-by-point responses to their comments. As part of the EMBO publication's Transparent Editorial Process, EMBO reports publishes online a Review Process File (RPF) to

accompany accepted manuscripts. This File will be published in conjunction with your paper and will include the referee reports, your point-by-point response and all pertinent correspondence relating to the manuscript.

<https://www.embopress.org/page/journal/14693178/authorguide#transparentprocess>

5) a complete author checklist, which you can download from our author guidelines

<https://www.embopress.org/page/journal/14693178/authorguide>. Please insert information in the checklist that is also reflected in the manuscript. The completed author checklist will also be part of the RPF.

6) Please note that all corresponding authors are required to supply an ORCID ID for their name upon submission of a revised manuscript (<<https://orcid.org/>>). Please find instructions on how to link your ORCID ID to your account in our manuscript tracking system in our Author guidelines

<<https://www.embopress.org/page/journal/14693178/authorguide#authorshipguidelines>>

Additional information on source data and instruction on how to label the files are available:

<https://www.embopress.org/page/journal/14693178/authorguide#sourcedata>

9) Our journal encourages inclusion of *data citations in the reference list* to directly cite datasets that were re-used and obtained from public databases. Data citations in the article text are distinct from normal bibliographical citations and should directly link to the database records from which the data can be accessed. In the main text, data citations are formatted as follows: "Data ref: Smith et al, 2001" or "Data ref: NCBI Sequence Read Archive PRJNA342805, 2017". In the Reference list, data citations must be labeled with "[DATASET]". A data reference must provide the database name, accession number/identifiers and a resolvable link to the landing page from which the data can be accessed at the end of the reference. Further instructions are available at <http://www.embopress.org/page/journal/14693178/authorguide#referencesformat>

10) Regarding data quantification (see Figure Legends:

<https://www.embopress.org/page/journal/14693178/authorguide#figureformat>)

11) The journal requires a statement specifying whether or not authors have competing interests (defined as all potential or actual interests that could be perceived to influence the presentation or interpretation of an article). In case of competing

interests, this must be specified in your disclosure statement. Further information: <https://www.embopress.org/competing-interests>

12) Please also note our reference format:

I look forward to seeing a revised version of your manuscript when it is ready. Please let me know if you have questions or comments regarding the revision.

Kind regards,

Deniz Senyilmaz Tiebe

Deniz Senyilmaz Tiebe, PhD
Scientific Editor
EMBO Reports

Referee #1:

Ham and colleagues present a succinct study detailing the impact of two genes linked to Wolfram Syndrome, WFS1 and CISD2, have on IP3R-related calcium release from the ER. Loss of these genes independently reduces IP3R activity and in flies causes 'diabetic' phenotypes (increased circulating glucose, defective response to sugar intake and insulin-like peptide expression). They show that WFS1/CISD2 overexpression can independently restore IP3R activity, and, excitingly, show this can be achieved with a small peptide of CISD2/dCISD. This is a focussed, well conducted study, that makes a few clear points indicating the relationship between WFS1/CISD2 and IP3R in vitro and in vivo, and show translatable relevance to in vivo pathology. I am not an expert in the background disease context, but the presentation of the Discussion indicates that ER calcium dysregulation is appreciated in WS but the mechanisms linking WFS1/CISD2 were not clear, so this study provides a valuable contribution. As the experimental conditions appear well controlled and executed, I have only relatively minor comments:

Specific comment:

Can the authors show that the different short peptides are being expressed equally (or at all) in the cell and fly systems, at the peptide level rather than the mRNA level. Dot blots targeting the Flag tags would be applicable here.

I believe that the C101 is one of the residues that coordinates the FeS cluster binding. Do the authors think the FeS cluster plays part of the mechanism? What about other FeS cluster binding site residues - do they have a specific impact on rescue?

What about a possible role of CISD1 here? The peptide sequence is almost identical in CISD1. Could there be some cross-talk?

Minor

- It would be good to know clearly what the data points on the quantification charts represent: independent wells or cell preparations. The n values indicated in the legend must be indicated ~100 cells. Is this across ~9 preps?
- Fig 2C. the labelling is a little confusing. The authors use "HA-WFS1 FL" at the side label for all constructs but this is only correct for what is labelled "WT" above. Swapping the WT for FL would correct this.
- Fig 2E. The authors noted looking at evolutionarily-conserved AAs. I'd be curious to know what species they compared and see which ones they focussed on.
- Fig 2F,G. Presumably, the dose-dependent expression is on top of endogenous, so could be referred to as dose-dependent 'overexpression'.
- Fig. 4. It would be useful to know the relative levels of dWFS1 and dCISD transgenic expression here compared to endogenous.
- Fig 5C-E. The label for WFS1 KO + WT CISD2 peptide is misspelled.
- It would make sense to have the EV5 (discussed in the text before EV4) as EV4.
- The authors should clarify where they think dCISD resides. They have recently published that dCISD is a OMM protein while the final depiction in 6G, which only specifically reflects CISD2, implies it should be on ER. DO they think dCISD is on both ER and OMM?
- I was wondering about the relative binding of WFS1 and CISD2 but I am confused about the data in EV7C. The Flag blots for the IP are quite different from the WCL. This IP is not convincing to me.
- Have the authors explored using cell-permeable peptides?

Referee #2:

In their manuscript, Ham et al. demonstrate the potential reciprocal rescue between two Wolfram syndrome causative genes, WFS1 and CISD2. They suggest that the peptide derived from CISD2 could alleviate IP3R function and associated diabetic symptoms.

The work presents potentially significant findings; however, there are several points requiring experimental clarification:

1. SERCA Function and Ca Uptake: The results indicating unaltered SERCA function and Ca uptake are not entirely convincing. The free Ca used in the uptake assay (not given in Methods but probably around pCa6), likely at an unphysiological concentration, may fully saturate the ER Ca uptake machinery, potentially masking any differences.
2. IP3R Overexpression in Drosophila: While IP3R overexpression in Wfs1 or Cisd2 Drosophila corrected the diabetic phenotype, it is also essential to show that IP3R overexpression corrects the Ca homeostasis (similarly to experiments depicted in Figure 3).
3. Cysteine 101 Mutation in CISD2: The mutation of cysteine 101 in CDGSH iron sulfur domain of CISD2 affects protein's function and probably also its structure. Previous research has indicated its role in interactions with BCL2. Loss of coimmunoprecipitation with IP3R does therefore not definitively establish this as the binding region.
4. Immunoprecipitation Experiments: The IP experiments in Fig 2 are all conducted under overexpression conditions. While the CISD2-IP3R interaction has been previously reported, the novel WFS1-IP3R interaction should also be validated using endogenous proteins.
5. 18aa Peptide Functionality: While the 18aa peptide is fully protective in the WFS1 and CISD2 deficient models, two concerns arise:
 - The authors do not provide evidence of the peptide's binding to IP3R. Given the potential interaction with endogenous CISD2, which forms dimers, there is a possibility that its function may be influenced or that it interacts with other target proteins. Figure 5B is aimed to show that the peptide blocks the IP3R-CISD2 interaction, but it could also be inferred that it directly interacts with CISD2, thus hindering its interaction with IP3R.
 - The absence of a peptide-only group in the rescue experiments depicted in Figure 5 limits the interpretability of the data. The peptide appears to increase ER Ca release by 40 units in control cells (Figure EV5A) and by a similar magnitude in CISD2 deficient cells (Figure 5F). This suggests that the peptide's effect is not influenced by CISD2 levels, contrary to expectations. It would help to have conditions in same experiment to understand the situation

Addressing these experimental points would strengthen the conclusions drawn from the manuscript and provide a more robust foundation for the proposed mechanisms.

Point-by-point responses to the reviewers

Our responses to the reviewers' questions are shown in blue (see below).

Referee #1:

Ham and colleagues present a succinct study detailing the impact of two genes linked to Wolfram Syndrome, WFS1 and CISD2, have on IP3R-related calcium release from the ER. Loss of these genes independently reduces IP3R activity and in flies causes 'diabetic' phenotypes (increased circulating glucose, defective response to sugar intake and insulin-like peptide expression). They show that WFS1/CISD2 overexpression can independently restore IP3R activity, and, excitingly, show this can be achieved with a small peptide of CISD2/dCISD. This is a focussed, well conducted study, that makes a few clear points indicating the relationship between WFS1/CISD2 and IP3R in vitro and in vivo, and show translatable relevance to in vivo pathology. I am not an expert in the background disease context, but the presentation of the Discussion indicates that ER calcium dysregulation is appreciated in WS but the mechanisms linking WFS1/CISD2 were not clear, so this study provides a valuable contribution. As the experimental conditions appear well controlled and executed, I have only relatively minor comments:

Specific comment:

Can the authors show that the different short peptides are being expressed equally (or at all) in the cell and fly systems, at the peptide level rather than the mRNA level. Dot blots targeting the Flag tags would be applicable here.

> Following the reviewer's suggestion, we performed dot blot analysis to assess the protein expression levels of the various peptides used in our study. The results showed that the WT human CISD2 peptide (N'-CRCWRSKTFPACDGSHNK-C'), C101A human CISD2 peptide (N'-CRAWRSKTFPACDGSHNK-C'), 16 amino acid human CISD peptide (N'-CRCWRSKTFPACDGSH-C'), and 14 amino acid human CISD peptide (N'-CWRSKTFPACDGSH-C') were expressed at similar levels in cells (A). Additionally, UAS-dCISD peptide line flies were found to express the dCISD peptide as expected (B). Hence, we further confirmed, that all of the CISD peptides used in our paper were expressed properly at protein levels through these additional experiments. We have added this new data in Expanded View Figure 10B and 13B.

<<Expression of human CISD2 peptides and *Drosophila* CISD peptide. (A) HEK293T cells were transfected as indicated and cell lysates were subjected to anti-Flag immunoprecipitation followed by dot blot analysis. (B) Fly whole body lysates were subjected to anti-Flag immunoprecipitation followed by dot blot analysis.>>

I believe that the C101 is one of the residues that coordinates the FeS cluster binding. Do the authors think the FeS cluster plays part of the mechanism? What about other FeS cluster binding site residues - do they have a specific impact on rescue?

> As the reviewer suggested, we observed whether other residues in the Fe-S cluster binding region of CISD2, C99, C110 and H114 affect its physical interaction with IP₃R1. We performed co-immunoprecipitation assay using IP₃R1 and CISD2 mutants of which each of the four residues in the Fe-S cluster binding region was substituted with alanine. While the CISD2 C101A mutant lost its binding ability towards IP₃R1, CISD2 C99A, CISD2 C110A, and CISD2 H114A mutants retained their binding abilities (A). Thus, we confirmed that residues related to the Fe-S cluster binding region of CISD2, except for

cysteine 101, do not affect its protein-protein interaction with IP₃R1 leading us to conclude that CISD2 interacts with IP₃R1 independently of its Fe-S cluster binding function. This data has been added to Expanded View Figure 7B.

What about a possible role of CISD1 here? The peptide sequence is almost identical in CISD1. Could there be some cross-talk?

> Following the reviewer's suggestion, we decided to determine whether there exists a cross-talk between CISD1 and CISD2. Given that our previous paper has verified that CISD1 regulates IP₃R, we investigated if CISD1 affects ER calcium release in the absence of CISD2 (Ham *et al.*, 2023). When CISD1 was overexpressed, ER calcium release increased compared to control in WT cells, consistent with our previous results (Ham *et al.*, 2023). Overexpression of WT CISD1 redeemed the reduced ER calcium release in CISD2 lacking cells (A). Given that the CISD2 peptide sequence is akin to the corresponding sequence of CISD1, we generated the CISD1 peptide consisting of the following sequence: N'-CRCWRSKKFPFCDAHTK-C'. When CISD1 peptide was overexpressed, ER calcium release was elevated compared to control in WT cells. Also, overexpression of CISD1 peptide rescued the decreased ER calcium release in CISD2 lacking cells (B). Based on these results, we concluded that WT CISD1 or CISD1 peptide affects ER calcium release regardless of the presence of CISD2.

<<Measurement of ER calcium release in cells expressing CISD1 and CISD1 peptide. (A) Measurement of ER calcium modulation in WT HEK293 cells transfected with empty vector (black, n=84 cells, 14 coverslips) or CISD1 (blue, n=12 cells, 12 coverslips) and CISD2 KO HEK293 cells transfected with empty vector (green, n=95 cells, 13 coverslips) or CISD1 (purple, n=111 cells, 17 coverslips). (B) Measurement of ER calcium modulation in WT HEK293 cells transfected with empty vector (black, n=73 cells, 10 coverslips) or CISD1 peptide (blue, n=104 cells, 9 coverslips) and CISD2 KO HEK293 cells transfected with empty vector (green, n=86 cells, 13 coverslips) or CISD1 peptide (purple, n=113 cells, 11 coverslips). The right-side bar graphs indicate the quantification of the normalized calcium traces using AOC of calcium fluxes during ATP treatment. Data information: All figures are representatives of three or more independent experiments. All quantifications were analyzed by one-way ANOVA with Tukey multiple-comparison test. ** $p < 0.01$, **** $p < 0.0001$. All data are presented as mean \pm SD.>>

Minor

- It would be good to know clearly what the data points on the quantification charts represent: independent wells or cell preparations. The n values indicated in the legend must be indicated ~100 cells. Is this across ~9 preps?

> The data points on the quantification charts represent independent coverslips. Calcium measurements were conducted for at least 5 coverslips and 50 cells. As the reviewer suggested, we indicated the number of cells and coverslips in the figure legends for all of the calcium measurement experiments.

- Fig 2C. the labelling is a little confusing. The authors use "HA-WFS1 FL" at the side label for all constructs but this is only correct for what is labelled "WT" above. Swapping the WT for FL would correct this.

> We apologize for the confusion regarding the labelling of the constructs. We have made the necessary correction.

- Fig 2E. The authors noted looking at evolutionarily-conserved AAs. I'd be curious to know what species they compared and see which ones they focused on.

> We have compared evolutionarily-conserved amino acids sequence of CISD2 across *Homo sapiens* (NM_001008388.4), *Bos taurus* (NM_001080338.1), *Mus musculus* (NM_025902.3), *Xenopus laevis* (NM_001089751.1), *Danio rerio* (NM_200383.1), *Drosophila melanogaster* (NM_143427.4), and *Caenorhabditis elegans* (NM_001129176.1). To address the reviewer's suggestion, we have included a schematic representation of this comparison in the Expanded View Figure 7A for more clarity.

- Fig 2F, G. Presumably, the dose-dependent expression is on top of endogenous, so could be referred to as dose-dependent 'overexpression'.

> We agree with the reviewer's suggestion that referring to "dose-dependent expression" as "dose-dependent overexpression" more accurately reflects that it is in addition to the endogenous levels. Accordingly, we have revised the manuscript to incorporate this terminology.

- Fig. 4. It would be useful to know the relative levels of dWFS1 and dCISD transgenic expression here compared to endogenous.

> Following the reviewer's suggestion, we verified protein levels of dWFS1 and dCISD in *w1118*, *dWFS1* KO, and *dCISD* KO flies expressing UAS-*LacZ*, UAS-*dWFS1*, or UAS-*dCISD* with the *tub-Gal4* driver using immunoblotting. As a result, the relative protein levels of dWFS1 and dCISD in transgenic flies were determined to be approximately 7-fold and 3-fold higher, respectively, compared to the endogenous protein levels (A and B). We have added this new data in the Expanded View Figure 9.

<<Expression of *Drosophila* WFS1 and CISD. (A) Immunoblot analysis of dWFS1 in *w1118* and *dWFS1* KO flies expressing UAS-*LacZ* or UAS-*dWFS1*. (B) Immunoblot analysis of dCISD in *w1118* and *dCISD* KO flies expressing UAS-*LacZ* or UAS-*dCISD*. The upper band indicates Flag-tagged dCISD and the lower band indicates endogenous dCISD. The asterisk denotes the band of interest. Anti-tubulin blot was used as a loading control. Bottom bar graphs represent the band intensity of dWFS1 or dCISD, which was quantified using image J.>>

- Fig 5C-E. The label for WFS1 KO + WT CISD2 peptide is misspelled.

> We apologize for the labeling error. We have corrected it in the revised manuscript. We appreciate your careful review.

- It would make sense to have the EV5 (discussed in the text before EV4) as EV4.

> Following the reviewer's feedback, we have revised the manuscript to reorder the expanded view figures, placing EV5 before EV4, as EV5 is discussed earlier in the text. We believe this adjustment improves the logical flow of the presentation.

- The authors should clarify where they think dCISD resides. They have recently published that dCISD is a OMM protein while the final depiction in 6G, which only specifically reflects CISD2, implies it should be on ER. DO they think dCISD is on both ER and OMM?

> As the reviewer noted, our previous study showed that dCISD co-localizes with mitochondria when expressed in mammalian cells. Human CISD1 and CISD2 are conserved as a single protein in *Drosophila*, showing more than 74% protein sequence homology. Given that we have confirmed that dCISD shares conserved functions with hCISD2, we aimed to identify the specific organelle where dCISD localizes when expressed in fly tissues. We observed that dCISD localizes to both the ER and mitochondria in the fly fat body (A and A').

Furthermore, recent studies have proposed that hCISD1 and hCISD2 are mitochondria-associated membranes (MAMs) proteins, as they reside in both the ER and mitochondria (Lee *et al*, 2024).

Based on these findings, we suppose that dCISD performs functions analogous to both hCISD1 and hCISD2 in the fly system, acting as a single protein that localizes to both the mitochondria and the ER.

<<Localization of *Drosophila* CISD in fat body tissue of flies. (A) Representative images of fat body of third instar larvae expressing *tub>dCISD* and *ER-GCaMP*, stained with streptavidin and anti-Flag antibodies. Scale bars, 2 μ m. (A') Co-localization analysis using Pearson's correlation coefficient (n=9).>>

- I was wondering about the relative binding of WFS1 and CISD2 but I am confused about the data in EV7C. The Flag blots for the IP are quite different from the WCL. This IP is not convincing to me.

> We apologize for the confusion caused by incorrect labeling. The correct labels should be as follows: 'IP: Flag' and 'IB: Myc.' This correction clarifies that the immunoprecipitation was performed using anti-Flag antibodies, and the immunoblot was conducted with anti-Myc antibodies to detect the interaction. We made this labeling correction in the revised manuscript to avoid any further misunderstanding.

- Have the authors explored using cell-permeable peptides?

> In response to the reviewer's feedback, to generate a cell-permeable peptide form of WT CISD2 peptide, we selected a cell-penetrating peptide (CPP), penetratin (N'-RQIKIWFQNRRMKWKK-C'). We synthesized a C-terminal CPP by linking the penetratin sequence to the WT CISD2 peptide, separated by a GGGs linker sequence (N'-GGGS-C'). Additionally, we conjugated Rhodamine B to the CPP as a fluorescent tag to facilitate visualization (A). We verified successful cellular delivery of the synthesized WT CISD2-CPP by visualizing its uptake (B).

Furthermore, to investigate whether WT Cisd2-CPP could rescue disrupted calcium fluxes in Wfs1 or Cisd2 knockout cells, we measured ER calcium release. Incubation without (-) or with CPP alone did not impact ER calcium fluxes in WT, Wfs1 knockout, or Cisd2 knockout cells. However, incubation with WT Cisd2-CPP notably increased ER calcium release in WT cells and restored diminished ER calcium release in Wfs1 and Cisd2 knockout cells (C and D).

This result signifies that WT Cisd2 peptide, when delivered into cells as a cell-permeable peptide, effectively rescues impaired ER calcium flux in Wfs1- or Cisd2-deficient cells, in addition to its previously observed effect when overexpressed. As cell-penetrating peptides are an emerging and effective form of peptide drug, a cell-permeable Cisd2 peptide may be proposed as a potential therapeutic for Wolfram syndrome. This new data has been added to the Expanded View Figure 12.

<<Measurement of ER calcium release in cells treated with cell-permeable Cisd2 peptides. (A) Schematic diagram representing cell-penetrating peptide (CPP) and WT Cisd2-CPP. (B) Representative images of cells incubated with or without CPP and WT Cisd2-CPP. Peptide uptake was visualized via Rhodamine B fluorescence (red), transmitted-photomultiplier tube (T-PMT) as brightfield, and merged images. Scale bar, 10 μ m. (C) ER calcium modulation of WT HEK293 cells was measured after incubation with pure buffer (black, n=96 cells, 12 coverslips), 10 μ M CPP (blue, n=97 cells, 14 coverslips), or 10 μ M WT Cisd2-CPP (purple, n=92 cells, 12 coverslips) at 37°C for 1 hour. ER calcium modulation of Wfs1 KO cells was measured after incubation with pure buffer (orange, n=107 cells, 10 coverslips), 10 μ M CPP (red, n=95 cells, 10 coverslips), or 10 μ M WT Cisd2-CPP (pink, n=112 cells, 14 coverslips) at 37°C for 1 hour. (D) ER calcium modulation of WT HEK293 cells was measured after incubation with pure buffer (black, n=107 cells, 11 coverslips), 10 μ M CPP (blue, n=85 cells, 11 coverslips), or 10 μ M WT Cisd2-CPP (purple, n=83 cells, 14 coverslips) at 37°C for 1 hour. ER calcium modulation of Cisd2 KO cells was measured after incubation with pure buffer (green, n=84 cells, 14 coverslips), 10 μ M CPP (red, n=93 cells, 9 coverslips), or 10 μ M WT Cisd2-CPP (pink, n=117 cells, 11 coverslips) at 37°C for 1 hour. The right-side bar graphs indicate the quantification of the normalized calcium traces using AOC of calcium fluxes during ATP treatment. Data information: All figures are representatives of three or more independent experiments. All quantifications were analyzed by one-way ANOVA with Tukey multiple-comparison test. * $p < 0.05$, ** $p < 0.01$, **** $p < 0.0001$. ns, not significant. All data are presented as mean \pm SD.>>

Referee #2:

In their manuscript, Ham et al. demonstrate the potential reciprocal rescue between two Wolfram syndrome causative genes, WFS1 and CISD2. They suggest that the peptide derived from CISD2 could alleviate IP3R function and associated diabetic symptoms.

The work presents potentially significant findings; however, there are several points requiring experimental clarification:

1. SERCA Function and Ca Uptake: The results indicating unaltered SERCA function and Ca uptake are not entirely convincing. The free Ca used in the uptake assay (not given in Methods but probably around pCa6), likely at an unphysiological concentration, may fully saturate the ER Ca uptake machinery, potentially masking any differences.

> Intracellular and physiological calcium concentrations are generally within the range of approximately pCa4 to pCa8 (Linderman *et al*, 1990; Marban *et al*, 1987), and the calcium concentration of the buffer used in our experiment was calculated to be approximately pCa6.7. Additionally, according to previous studies, SERCA activity assays are typically conducted within a pCa5 to pCa8 range (Fajardo *et al*, 2013; Stroik *et al*, 2018; Zamoon *et al*, 2005).

To confirm whether the calcium concentration of our current buffer is suitable for accurately assessing SERCA activity, we measured ER calcium release using buffers with a higher calcium concentration (pCa6) and a lower calcium concentration (pCa8) than our standard buffer. Compared to our original buffer, the pCa6 buffer produced a more rapid increase in SERCA activity, while the pCa8 buffer showed reduced SERCA activity (A). These results indicate that SERCA activity was measured within a suitable calcium range that did not saturate ER calcium uptake. However, considering the reviewer's concern, we further analyzed SERCA activity using pCa8 buffer.

When we measured SERCA activity in WT, WFS1 knockout, and CISD2 knockout cells using the pCa8 buffer, no significant differences were observed among the cell lines, consistent with our previous results using pCa6.7 buffer (B).

Hence, we confirmed that loss of WFS1 or CISD2 does not affect SERCA activity in cells when tested at two different calcium concentrations, pCa6.7 and pCa8. This result assures that the calcium concentration of our standard buffer used in paper is suitable for measuring SERCA activity. Altogether, we consistently propose that CISD2 and WFS1 modulate IP₃R activity, but not SERCA activity.

<<Measurement of the SERCA activity using different calcium concentration buffers. (A) Measurement of the SERCA activity of WT HEK293 cells treated with pCa6 calcium loading buffer (black, n=80 cells, 10 coverslips), pCa6.7 calcium loading buffer (blue, n=69 cells, 10 coverslips), and pCa8 calcium loading buffer (purple, n=97 cells, 10 coverslips). (B) Measurement of the SERCA activity of WT HEK293 cells (black, n=73 cells, 8 coverslips), WFS1 KO HEK293 cells (orange, n=83 cells, 8 coverslips), and CISD2 KO HEK293 cells (green, n=84 cells, 8 coverslips) treated with pCa8 calcium loading buffer. The right-side bar graph represents the magnitude of the change during CaCl₂ treatment, indicated as gray in the left-side graph. Data information: All figures are representatives of three or more independent experiments. All quantifications were analyzed by one-way ANOVA with Tukey multiple-comparison test. *** $p < 0.001$, **** $p < 0.0001$. ns, not significant. All data are presented as mean \pm SD.>>

2. IP3R Overexpression in *Drosophila*: While IP3R overexpression in *Wfs1* or *Cisd2* *Drosophila* corrected the diabetic phenotype, it is also essential to show that IP3R overexpression corrects the Ca homeostasis (similarly to experiments depicted in Figure 3).

> Following the reviewer's suggestion, we measured calcium levels under IP₃R1 overexpression conditions. In WT cells, IP₃R1 overexpression led to increased ER calcium release and cytosolic calcium levels compared to control cells. Additionally, IP₃R1 overexpression in WFS1 or CISD2 knockout cells restored decreased ER calcium release and cytosolic calcium levels (A, B, C, and D).

Furthermore, we investigated calcium homeostasis in *dWFS1* and *dCISD* mutant flies. As expected, *IP₃R* transgenic flies exhibited increased ER calcium release and cytosolic calcium levels. Consistent with our cell experiment results, *dWFS1* and *dCISD* mutant flies showed reduced ER calcium release and cytosolic calcium levels compared to control flies, which were rescued by *IP₃R* overexpression (E, F, G, and H).

In both mammalian cells and *Drosophila*, the decrease in ER calcium release and cytosolic calcium levels induced by loss of WFS1 or CISD2 was restored to normal levels through IP₃R overexpression. This result suggests that the improved diabetes-like phenotypes observed in *dWFS1* or *dCISD* mutant flies overexpressing *IP₃R* were initially stemmed from the recovery of calcium homeostasis. This new data has been added to the Expanded View Figure 4 (calcium measurement in flies) and 5 (calcium measurement in mammalian cells).

<<Measurement of ER calcium release and cytosolic calcium levels in mammalian cell and *Drosophila* under IP₃R overexpression. (A) Measurement of ER calcium modulation in WT HEK293 cells transfected with empty vector (black, n=84 cells, 8 coverslips) or IP₃R1 (blue, n=104 cells, 8 coverslips) and WFS1 KO HEK293 cells transfected with empty vector (orange, n=107 cells, 8 coverslips) or IP₃R1 (purple, n=97 cells, 8 coverslips). (B) Measurement of cytosolic calcium modulation in WT HEK293 cells transfected with empty vector (black, n=99 cells, 7 coverslips) or IP₃R1 (blue, n=117 cells, 10 coverslips) and WFS1 KO HEK293 cells transfected with empty vector (orange, n=71 cells, 12 coverslips) or IP₃R1 (purple, n=84 cells, 7 coverslips). (C) Identical experiments measuring ER calcium modulation were conducted in WT HEK293 cells transfected with empty vector (black, n=108 cells, 8 coverslips) or IP₃R1 (blue, n=114 cells, 7 coverslips) and CISD2 KO HEK293 cells transfected with empty vector (green, n=80 cells, 5 coverslips) or IP₃R1 (purple, n=116 cells, 7 coverslips). (D) Identical experiments measuring cytosolic calcium modulation were conducted in WT HEK293 cells transfected with empty vector (black, n=82 cells, 7 coverslips) or IP₃R1 (blue, n=62 cells, 9 coverslips) and CISD2 KO HEK293 cells transfected with empty vector (green, n=54 cells, 10 coverslips) or IP₃R1 (purple, n=61 cells, 8 coverslips). (E, F) Measurement of ER calcium and cytosolic calcium modulations in control (*tub>LacZ*, black) and IP₃R transgenic flies (*tub>IP₃R*, blue). Similar experiments were also conducted in *dWFS1* mutant flies (*dWFS1 KO, tub>LacZ*, orange) and *dWFS1* null flies expressing IP₃R (*dWFS1 KO, tub>IP₃R*, purple). (G, H) Measurement of ER calcium and cytosolic calcium modulations in control (*tub>LacZ*, black) and IP₃R transgenic flies (*tub>IP₃R*, blue). Similar experiments were also conducted in *dCISD* mutant flies (*dCISD KO, tub>LacZ*, green) and *dCISD* null flies expressing IP₃R (*dCISD KO, tub>IP₃R*, purple). The right-side bar graphs indicate the quantification of the normalized calcium traces using AOC or AUC of calcium fluxes during ATP treatment (n=10). All quantifications were analyzed by one-way ANOVA with Tukey multiple-comparison test. * $p < 0.05$, ** $p < 0.01$, *** $p < 0.001$, **** $p < 0.0001$. All data are presented as mean \pm SD.>>

3. Cysteine 101 Mutation in CISD2: The mutation of cysteine 101 in CDGSH iron sulfur domain of CISD2 affects protein's function and probably also its structure. Previous research has indicated its role in interactions with BCL2. Loss of co-immunoprecipitation with IP3R does therefore not definitively establish this as the binding region.

> To investigate whether BCL2 affects the CISD2-IP₃R interaction, we performed a co-IP assay with CISD2 and IP₃R under BCL2 knockdown condition. We found that CISD2 maintained its ability to bind to IP₃R1 despite BCL2 knockdown indicating that CISD2 binds to IP₃R1 independently of BCL2 (A).

As the reviewer noted, the co-IP of the CISD2 C101A mutant with IP₃R1 alone does not ensure that CISD2 cysteine 101 is the binding site for IP₃R1. However, during the revision process, we performed additional experiments. As shown in the response to question 2 from reviewer #1, we found that mutations in various other residues critically associated with the iron-sulfur motif of CISD2 did not affect the binding between IP₃R1 and CISD2. Therefore, we concluded that the structure of CDGSH iron-sulfur binding sites does not play a crucial role in the interaction between CISD2 and IP₃R1.

Furthermore, to confirm the role of CISD2 Cys101 as the binding site, we conducted a co-IP assay with the WT CISD2 peptide, C101A CISD2 peptide, and IP₃R1. Since the peptide sequence does not form a full protein structure but contains the proposed binding region, this allowed us to assess the importance of Cys101 for binding. The WT CISD2 peptide directly interacted with IP₃R1, whereas the C101A CISD2 peptide did not, supporting the conclusion that cysteine 101 is a crucial residue in CISD2 for its interaction with IP₃R1 (B).

These results further confirm that the cysteine 101 residue of CISD2 is critical for the binding between CISD2 and IP₃R1. This new data has been added to the Expanded View Figure 11A and 14.

<<CISD2 binds to IP₃R1 via cysteine 101 independently of BCL2. (A) WT HEK293T cells were transfected as indicated and cell lysates were subjected to anti-Myc immunoprecipitation followed by immunoblot analysis. (B) WT and CISD2 KO HEK293 cells were transfected as indicated and cell lysates were subjected to anti-Flag immunoprecipitation followed by immunoblot or dot blot analysis. The asterisks denote the band of interest.>>

4. Immunoprecipitation Experiments: The IP experiments in Fig 2 are all conducted under overexpression conditions. While the CISD2-IP₃R interaction has been previously reported, the novel WFS1-IP₃R interaction should also be validated using endogenous proteins.

> Following the reviewer's feedback, we performed co-immunoprecipitation assay using endogenous proteins to verify the WFS1-IP₃R1 interaction. We observed the interaction between WFS1 and IP₃R1 at endogenous protein levels (A), thereby supporting our findings from the overexpression studies. We have included this new data in the Expanded View Figure 6.

<<Interaction between IP₃R1 and WFS1 at the endogenous level. (A) HEK293 cell lysates were subjected to anti-IgG or anti-WFS1 immunoprecipitation followed by immunoblot analysis. The asterisks denote the band of interest.>>

5. 18aa Peptide Functionality: While the 18aa peptide is fully protective in the WFS1 and CISD2 deficient models, two concerns arise:

-The authors do not provide evidence of the peptide's binding to IP₃R. Given the potential interaction with endogenous CISD2, which forms dimers, there is a possibility that its function may be influenced or that it interacts with other target proteins. Figure 5B is aimed to show that the peptide blocks the IP₃R-CISD2 interaction, but it could also be inferred that it directly interacts with CISD2, thus hindering its interaction with IP₃R.

> We understand the reviewer's insight that our result, shown in Figure 5B, is insufficient to provide evidence that WT CISD2 peptide directly binds to IP₃R. In response to the reviewer's feedback, we conducted co-IP experiments to determine whether CISD2 peptide directly interacts with IP₃R1 in WT and CISD2 KO HEK293 cells. We observed that WT CISD2 peptide bound to IP₃R1 in both WT and CISD2 KO HEK293 cells, whereas the C101A CISD2 peptide did not bind to IP₃R1 in either cell type (data shown in the response to question 3 from the reviewer #2).

As we have shown in Figure 5B, the binding between CISD2 and IP₃R1 gradually decreased with dose-dependent overexpression of WT CISD2 peptide. This result suggests that CISD2 and the WT CISD2 peptide bind to the same region on IP₃R1. Therefore, we have revised the manuscript to clarify that WT CISD2 and the CISD2 peptide bind to the same region on IP₃R1.

-The absence of a peptide-only group in the rescue experiments depicted in Figure 5 limits the interpretability of the data. The peptide appears to increase ER Ca release by 40 units in control cells (Figure EV5A) and by a similar magnitude in CISD2 deficient cells (Figure 5F). This suggests that the peptide's effect is not influenced by CISD2 levels, contrary to expectations. It would help to have conditions in same experiment to understand the situation

> Following the reviewer's suggestion, we compared WT CISD2 peptide's effect on ER calcium release in WT and CISD2 knockout cells in the same experiment. When WT CISD2 peptide was overexpressed, ER calcium release increased by approximately 40 units both in WT and CISD2 knockout cells (A). Also, cytosolic calcium release was elevated by similar levels in WT and CISD2 lacking cells (B). Therefore, these results demonstrated that the effect of the WT CISD2 peptide is independent of CISD2 presence. Additionally, as mentioned in our response to question 3 from the reviewer #2, the WT CISD2 peptide directly binds to IP₃R1.

Altogether, CISD2 peptide can increase IP₃R activity in both WT and CISD2 KO cells because CISD2 peptide directly interacts with IP₃R. CISD2 and CISD2 peptide share the same binding region on IP₃R1, and they exhibit identical effects in enhancing IP₃R activity. Consequently, overexpression of either CISD2 or CISD2 peptide shows an additive increase in ER calcium release and cytosolic calcium levels in both WT and CISD2 KO cells. This new data has been added to the Expanded View Figure 8D and E.

<<Measurement of ER calcium release and cytosolic calcium levels in cells expressing WT CISD2 peptide. (A) Measurement of ER calcium modulation in WT HEK293 cells transfected with empty vector (black, n=129 cells, 14 coverslips) or WT CISD2 peptide (blue, n=117 cells, 14 coverslips) and CISD2 KO HEK293 cells transfected with empty vector (green, n=146 cells, 13 coverslips) or WT CISD2 peptide (purple, n=133 cells, 14 coverslips). (B) Measurement of cytosolic calcium modulation in WT HEK293 cells transfected with empty vector (black, n=119 cells, 10 coverslips) or WT CISD2 peptide (blue, n=98 cells, 11 coverslips) and CISD2 KO HEK293 cells transfected with empty vector (green, n=113 cells, 12 coverslips) or WT CISD2 peptide (purple, n=119 cells, 9 coverslips). The right-side bar graphs indicate the quantification of the normalized calcium traces using AOC or AUC of calcium fluxes during ATP treatment. Data information: All figures are representatives of three or more independent experiments. All quantifications were analyzed by one-way ANOVA with Tukey multiple-comparison test. ** $p < 0.01$, *** $p < 0.001$, **** $p < 0.0001$. All data are presented as mean \pm SD.>>

Addressing these experimental points would strengthen the conclusions drawn from the manuscript and provide a more robust foundation for the proposed mechanisms.

We appreciate your careful review and insight regarding the proposed mechanisms. We have provided detailed answers and experimental results to the questions in a point-by-point format. We hope our answers address the raised concerns and questions. Additionally, most of the requested data have been included in the revised manuscript. We believe this strengthens and consolidates the conclusions derived from this study.

References

- Fajardo VA, Bombardier E, Vigna C, Devji T, Bloemberg D, Gamu D, Gramolini AO, Quadrilatero J, Tupling AR (2013) Co-expression of SERCA isoforms, phospholamban and sarcolipin in human skeletal muscle fibers. *PLoS One* 8: e84304
- Ham SJ, Yoo H, Woo D, Lee DH, Park KS, Chung J (2023) PINK1 and Parkin regulate IP(3)R-mediated ER calcium release. *Nat Commun* 14: 5202
- Lee A, Sung G, Shin S, Lee SY, Sim J, Nhung TTM, Nghi TD, Park SK, Sathieshkumar PP, Kang I *et al* (2024) OrthoID: profiling dynamic proteomes through time and space using mutually orthogonal chemical tools. *Nat Commun* 15: 1851
- Linderman JJ, Harris LJ, Slakey LL, Gross DJ (1990) Charge-coupled device imaging of rapid calcium transients in cultured arterial smooth muscle cells. *Cell Calcium* 11: 131-144
- Marban E, Kitakaze M, Kusuoka H, Porterfield JK, Yue DT, Chacko VP (1987) Intracellular free calcium concentration measured with ¹⁹F NMR spectroscopy in intact ferret hearts. *Proc Natl Acad Sci U S A* 84: 6005-6009
- Stroik DR, Yuen SL, Janicek KA, Schaaf TM, Li J, Ceholski DK, Hajjar RJ, Cornea RL, Thomas DD (2018) Targeting protein-protein interactions for therapeutic discovery via FRET-based high-throughput screening in living cells. *Sci Rep* 8: 12560
- Zamoon J, Nitu F, Karim C, Thomas DD, Veglia G (2005) Mapping the interaction surface of a membrane protein: unveiling the conformational switch of phospholamban in calcium pump regulation. *Proc Natl Acad Sci U S A* 102: 4747-4752

Dear Prof. Chung,

Thank you for submitting your revised manuscript. It has now been seen by both of the original referees.

As you can see, referees find that the study is significantly improved during revision and recommend publication. However, I need you to address the points below before I can accept the manuscript.

- Please address the remaining concern of referee #1 (textually) by toning down the conclusions regarding the subcellular localization of dCISD.
- We note that there are currently 15 EV figures, which is more than we can technically accommodate. Therefore, 10 of these should be compiled in an appendix file, which is a PDF file with the figures labelled "Appendix Figure S1" etc. and the legends included underneath each figure. The appendix should have a table of contents with page numbers. Please remember to update the callouts in the manuscript text accordingly.
- Please remove the 'Author Contributions' section from the manuscript text.
- All research articles submitted as revised versions must include a structured methods section that includes a Reagents and Tools Table followed by a Methods and Protocols section. Please see <https://www.embopress.org/page/journal/14693178/authorguide#structuredmethods> for further information.
- "Materials and Methods" should be renamed as "Methods".
- Panel A should be removed for Fig EV6 and EV14 as there is only one panel.
- Please rename the figure panels with ' - i.e. A, A' should be A, B instead.
- Our production/data editors have asked you to clarify several points in the figure legends:
 - o Please note that the exact p values are not provided in the legends of figures 1A-G, H'; 2C, D, E; 3A-F; 4A-H, I'; 5B-H; 6B-E, F'; EV2 A-E, F'; EV4 A-D; EV5 A-D; EV8 A-E; EV10 C, EV11 B, C; EV12 C, D.
- Papers published in EMBO Reports include a 'synopsis' and 'bullet points' to further enhance discoverability. Both are displayed on the html version of the paper and are freely accessible to all readers. The synopsis includes a short standfirst summarizing the study in 1 or 2 sentences (max 35 words) that summarize the paper and are provided by the authors and streamlined by the handling editor. I would therefore ask you to include your synopsis blurb and 3-5 bullet points listing the key experimental findings.
- In addition, please provide an image for the synopsis. This image should provide a rapid overview of the question addressed in the study but still needs to be kept fairly modest since the image size cannot exceed 550 (width) x 300-600 (height) pixels.

Thank you again for giving us to consider your manuscript for EMBO Reports, I look forward to your minor revision.

Kind regards,

Deniz Senyilmaz Tiebe

--

Deniz Senyilmaz Tiebe, PhD
Senior Scientific Editor
EMBO Reports

Referee #1:

The authors have done a good job of providing substantial new data to address the points that I raised previously. These largely answer my previous concerns. The only aspect that I find is still unconvincing, or at least unclear, is the sub-cellular localisation of dCISD. The resolution of the presented images is too low to be very confident. However, this is not a critical piece of the story and these data are not included in the manuscript so I don't think it's essential to pursue it further here.

My final comment is that the cell-permeable peptide data are very nice and I think the authors could have made more of highlighting this e.g., in the abstract.

Referee #2:

The authors have addressed my comments comprehensively, and I am satisfied with the revisions. I have no further comments and support the acceptance of the paper.

Point-by-point responses to the reviewers

Our responses to the reviewers' questions are shown in blue (see below).

Referee #1:

The authors have done a good job of providing substantial new data to address the points that I raised previously. These largely answer my previous concerns. The only aspect that I find is still unconvincing, or at least unclear, is the sub-cellular localisation of dCISD. The resolution of the presented images is too low to be very confident. However, this is not a critical piece of the story and these data are not included in the manuscript so I don't think it's essential to pursue it further here.

> As the reviewer noted, the sub-cellular localization of dCISD was not mentioned or described in the original manuscript, so we did not make any changes in manuscript. However, we acknowledge the limitation in determining the sub-cellular localization of dCISD due to the low image resolution. Further studies with higher-resolution imaging techniques in fly fat body tissues will be necessary to precisely clarify this aspect.

My final comment is that the cell-permeable peptide data are very nice and I think the authors could have made more of highlighting this e.g., in the abstract.

> We appreciate the positive comments on the cell-permeable peptide data. As suggested, we have revised the abstract to better highlight this aspect.

Referee #2:

The authors have addressed my comments comprehensively, and I am satisfied with the revisions. I have no further comments and support the acceptance of the paper.

> We sincerely appreciate the reviewer's positive feedback and support for the acceptance of our manuscript.

Prof. Jongkyeong Chung
Seoul National University
School of Biological Sciences
1 Gwanak-ro, Gwanak-gu
Building 105 West Room 533
Seoul 08826
Korea, Republic of

Dear Prof. Chung,

Thank you for submitting your revised manuscript. I have now looked at everything and all is fine. Therefore, I am very pleased to accept your manuscript for publication in EMBO Reports.

Congratulations on a nice work!

Kind regards,

Deniz Senyilmaz Tiebe

--

Deniz Senyilmaz Tiebe, PhD
Senior Scientific Editor
EMBO Reports

--
